# Alpha 7 nicotinic acetylcholine receptors signaling boosts cell-cell interactions in macrophages effecting anti-inflammatory and organ protection

Yasuna Nakamura [1,10], Hirotaka Matsumoto [2,10], Chia-Hsien Wu[1], Daichi Fukaya[3], Rie Uni[4], Yosuke Hirakawa[5], Mikako Katagiri [6], Shintaro Yamada[6,7], Toshiyuki Ko[6,7], Seitaro Nomura[6,7], Youichiro Wada[8], Issei Komuro[9], Masaomi Nangaku [5], Reiko Inagi[4] & Tsuyoshi Inoue [1✉]

Activation of the cholinergic anti-inflammatory pathway (CAP) via vagus nerve stimulation has been shown to improve acute kidney injury in rodent models. While alpha 7 nicotinic acetylcholine receptor (α7nAChR) positive macrophages are thought to play a crucial role in this pathway, their in vivo significance has not been fully understood. In this study, we used macrophage-specific α7nAChR-deficient mice to confirm the direct activation of α7nAChRs in macrophages. Our findings indicate that the administration of GTS-21, an α7nAChR-specific agonist, protects injured kidneys in wild-type mice but not in macrophage-specific α7nAChR-deficient mice. To investigate the signal changes or cell reconstructions induced by α7nAChR activation in splenocytes, we conducted single-cell RNA-sequencing of the spleen. Ligand-receptor analysis revealed an increase in macrophage-macrophage interactions. Using macrophage-derived cell lines, we demonstrated that GTS-21 increases cell contact, and that the contact between macrophages receiving α7nAChR signals leads to a reduction in TNF-α. Our results suggest that α7nAChR signaling increases macrophage-macrophage interactions in the spleen and has a protective effect on the kidneys.

[1] Department of Physiology of Visceral Function and Body Fluid, Nagasaki University Graduate School of Biomedical Sciences, Nagasaki, Japan. [2] School of Information and Data Sciences, Nagasaki University, Nagasaki, Japan. [3] Department of Nephrology, Saitama Medical University, Saitama, Japan. [4] Division of CKD pathophysiology, The University of Tokyo Graduate School of Medicine, Tokyo, Japan. [5] Division of Nephrology and Endocrinology, The University of Tokyo Graduate School of Medicine, Tokyo, Japan. [6] Department of Cardiovascular Medicine, Graduate School of Medicine, the University of Tokyo, Tokyo, Japan. [7] Genome Science Division, Research Center for Advanced Science and Technology, The University of Tokyo, Tokyo, Japan. [8] Isotope Science Center, The University of Tokyo, Tokyo, Japan. [9] Department of Cardiovascular Medicine the University of Tokyo Graduate School of Medicine, Tokyo, Japan. [10] These authors contributed equally: Yasuna Nakamura, Hirotaka Matsumoto. ✉email: ts-inoue@nagasaki-u.ac.jp

The cholinergic anti-inflammatory pathway (CAP) was proposed as a neural-mediated immune control mechanism by Tracey et al. in 2002[1–3]. Initially, vagus nerve stimulation (VNS) was thought to suppress systemic tumor necrosis factor alpha (TNF-α) and cytokine production to elicit an anti-inflammatory response. In vitro experiments using antisense oligonucleotides on primary cultures of human macrophages identified the mediating role of alpha 7 nicotinic acetylcholine receptor (α7nAChR)[4]. VNS has been shown to ameliorate various inflammatory diseases, including sepsis[5,6], rheumatoid arthritis[7,8], gastrointestinal diseases (including Chron's disease and inflammatory bowel diseases)[9–12], lung disease[13], and acute kidney injury (AKI)[14–16].

AKI is defined as the sudden loss of kidney function, which can result in increased serum creatinine levels or decreased urine output. It can be caused by a variety of factors including renal hypoperfusion, surgery, sepsis, shock, and exposure to nephrotoxic substances. AKI is estimated to occur in one out of every five hospitalized patients[17]. As kidneys are responsible for fluid homeostasis, loss of function can be life-threatening and may require renal replacement therapies, such as kidney transplantation or hemodialysis, which can contribute significantly to the medical burden. Furthermore, a history of AKI increases the risk of chronic kidney disease (CKD)[18]. Despite its prevalence and severity, there is currently no established treatment for AKI, and appropriate fluid management is the only available intervention.

The vagus nerve stimulator, an electrical device developed for treating epilepsy and refractory depression, has been approved by the US Food and Drug Administration (FDA) and shown to be effective in clinical trials for inflammatory diseases such as rheumatoid arthritis[19] and Crohn's disease[9]. In addition, non-invasive vagus nerve stimulators like transcutaneous auricular vagus nerve stimulators have been developed, which do not require electrode implantation, and are gaining attention from researchers for their clinical applications[20–26]. Given these developments, VNS is a promising neuromodulatory treatment for AKI. The current assumed pathway of the CAP involves[2,27] afferent vagus nerve stimulation, followed by signals transmitted to the parasympathetic nucleus in the medulla oblongata, which are projected to the efferent vagus nerve and transmitted via the celiac ganglion to the spleen. Noradrenaline released from splenic nerve endings activates CD4 + T cells (CD4+, CD44high, and CD62Llow cells) which have β2 adrenergic receptors. These CD4 + T cells express acetylcholine synthase, namely choline acetyltransferase (ChAT), and produce acetylcholine (ACh)[27], which is then received by α7nAChR positive macrophages[4], resulting in systemic anti-inflammatory effects[28]. This response in the spleen could be considered a "non-neuronal cholinergic pathway" in that ACh is not received directly from nerve endings but delivered via immune cells such as CD4 + T cells. One recently reported pathway does not involve the spleen and occurs, instead, in organs with vagal innervation, such as the lung and guts[13]. In these organs, ACh released from vagus nerve endings is thought to act directly on the organ and produce an anti-inflammatory effect. Moreover, since the parasympathetic innervation of the kidney has been controversial[29], it is currently believed that spleen-mediated CAP is the primary route of action for renal protection.

Regarding the intracellular signaling pathways, experiments have shown that, after binding, α7nAChR inhibits nuclear translocation of nuclear factor kappa-light-chain-enhancer of activated B cells (NF-κB)[30–34], suppresses pyrin domain-containing 3 (NLRP3) inflammasome activation[35], and activates the Janus kinase 2 (JAK2)/signal transducer and activator of the transcription-3 (STAT3) pathway[36,37]. Recently, the hairy and enhancer of split-1 (Hes1), a transcriptional modulator, was also identified in macrophages[15]. Moreover, we previously reported that VNS ameliorates renal bilateral ischemia-reperfusion injury (IRI) in addition to systemic inflammation[14,15]. Ultrasound has also been shown to improve systemic inflammation, renal bilateral IRI kidney damage, and VNS by activating splenic CAP[38,39]. Since splenectomy abolishes the renoprotective effect and the transfer of VNS-treated splenocytes shows renoprotective effects even in naive wild-type (WT) mice[14], we believe that CAP is activated via the spleen. Furthermore, we previously demonstrated that the phenotype of macrophages migrating to the injured kidney changes to the M2 (anti-inflammatory) type[14] and that in cisplatin-induced AKI, a decrease in the number of macrophages migrating to the injured kidney due to a decrease in chemokine CCL2 leads to kidney protection[16].

The α7nAChR is considered important in the CAP for the following reasons: (1) In addition to electrical VNS, the administration of nicotine- or α7nAChR-specific agonists also protects the kidneys from renal IRI[40,41] and other AKIs[42,43]. (2) Global α7nAChR-deficient mice lack the anti-inflammatory and kidney protective effects of VNS[4,14,40]. (3) The adoptive transfer of VNS- or nicotine-treated macrophages to WT mice reduces renal damage from renal IRI injury[15]. The kidney protective effects of VNS or nicotinic acetylcholine receptor agonists have been confirmed in many rodent models, including renal bilateral IRI[14,15,39,40], LPS induces septic AKI[41,44], and cisplatin induces nephropathy[16,42], and is thought to involve the activation of splenic CAP, but no research has shown that it is mediated by α7nAChR on macrophages in vivo.

In contrast, the spleen and secondary lymphoid tissue contain various immune cells, including T and B lymphocytes, dendritic cells (DC), and macrophages. These immune cells also express nicotinic acetylcholine receptors, including the α7 subunit and muscarinic acetylcholine receptors[45–47], and several studies have reported that they produce immune responses via α7nAChRs. For example, 3-(2,4-Dimethoxybenzylidene)- anabaseine dihydrochloride (GTS-21), α7nAChR specific agonists, suppressed CD4+ regulatory T and effector T cell development and cytokine production (such as IL-2, IFN-γ, and IL-6) in an ovalbumin-induced antigen processing-dependent manner[48]. Furthermore, adoptively transferred B cells treated with α7nAChR antagonists showed decreased antibody production[49]. Moreover, there are some reports on the relationship between CAP activation and DCs in rodent models[50]. In a collagen-induced arthritis model, GTS-21 treatment also improved arthritis by suppressing DC differentiation[51]. These reports suggest that immune cells other than macrophages may also be affected by GTS-21 in the spleen.

We hypothesize that if the activation of splenic CAP occurs via α7nAChR on macrophages, then GTS-21 should have no anti-inflammatory effect in macrophage-specific α7nAChR knockout (KO) mice. We believe that this reaction occurs via α7nAChR on splenic macrophages, but since α7nAChRs are also present in the nervous system and skeletal muscles, autonomic ganglia[52], as well as other splenocytes such as lymphocytes[53], monocytes[53,54] and dendritic cells[54], we need to exclude the influence of α7nAChR on cells except for macrophages. Therefore, in this study, we attempted to elucidate the function of the α7nAChR in macrophages in vivo using newly generated macrophage-specific α7nAChR-deficient mice[55,56].

## Results

### α7nAChR stimulation ameliorates LPS-induced systemic inflammation and kidney injury in WT mice.
To eliminate the influence of various other in vivo factors and specifically target α7nAChR, we used GTS-21, a selective agonist of α7nAChR, in this experiment. Our first objective was to determine the

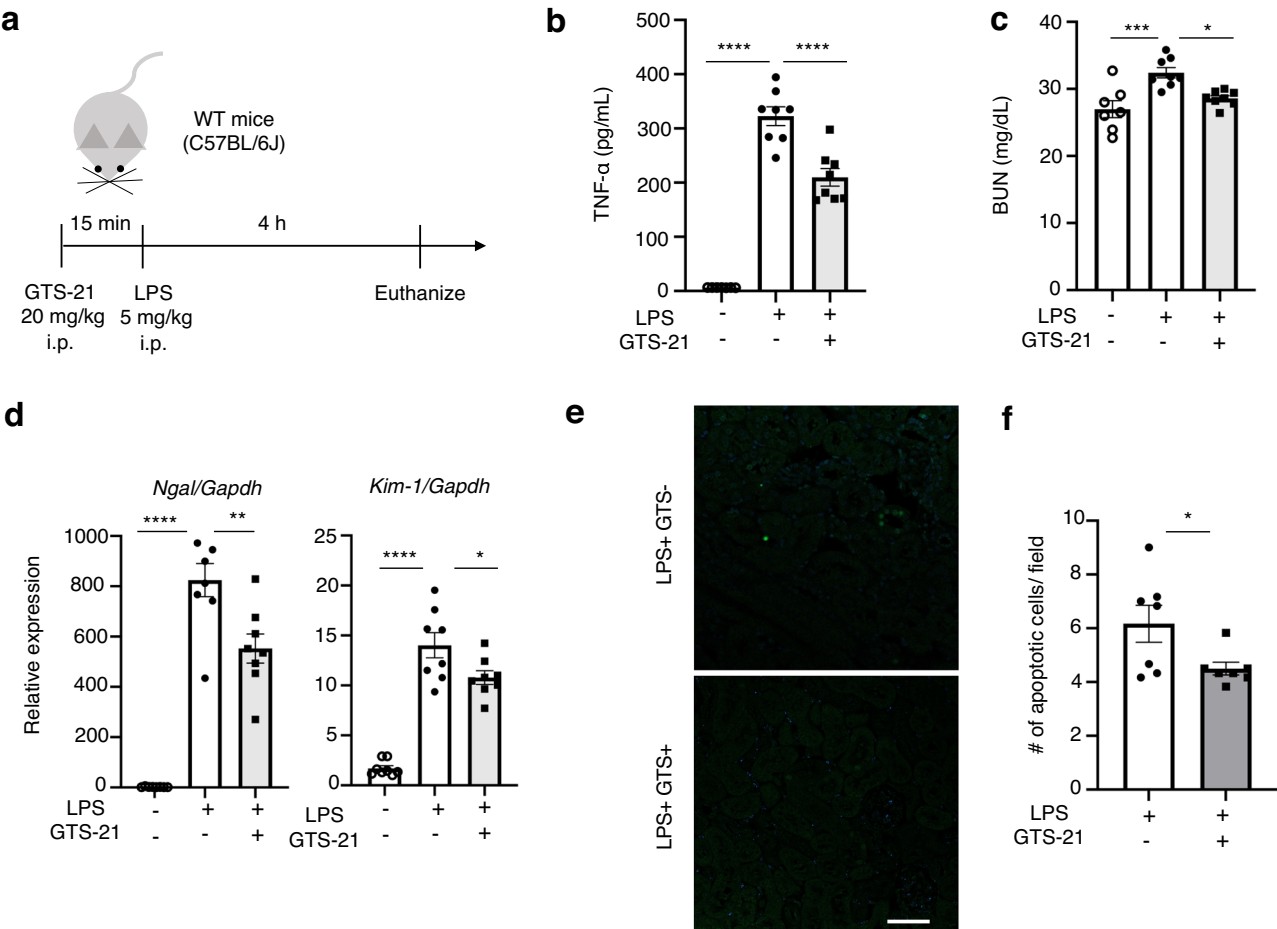

**Fig. 1 GTS-21 attenuates LPS-induced systemic inflammation and early kidney injury in WT mice. a** Experimental design: WT mice received 5 mg kg⁻¹ LPS intraperitoneal injection 15 min after GTS-21 (20 mg kg⁻¹) single dose injection ($n = 7$–8 of each group). Four hours later, they were euthanized, and blood and kidney samples were collected. **b** Plasma TNF-α level. **c** Plasma BUN. **d** RNA was extracted from whole kidney samples, and qPCR was performed. *Ngal* and *Kim-1*, which are marker genes of AKI, were measured. **e** Representative pictures of TUNEL staining and (**f**) number of apoptotic cells following LPS and GTS-21 administration. Scale bar = 50 µm. *$P < 0.05$, **$P < 0.01$, ***$P < 0.001$, ****$P < 0.0001$ (one-way ANOVA followed by Tukey's post hoc test (**b**–**e**) and Unpaired *t* test (**f**)). All data are presented as mean ± SEM. WT wild-type, GTS-21, 3-(2,4-Dimethoxybenzylidene)- anabaseine dihydrochloride, LPS lipopolysaccharide, PAS periodic acid-Schiff, TNF-α tumor necrosis factor α, BUN blood urea nitrogen, qPCR quantitative polymerase chain reaction, TUNEL Terminal deoxynucleotidyl transferase dUTP nick end labeling, ANOVA analysis of variance, SEM standard error of the mean.

effectiveness of GTS-21 against sepsis-induced AKI. Based on previous reports, which showed that pre-treatment with VNS improves AKI and protects kidneys from renal IRI[14,15], we administered GTS-21 before injecting LPS (Fig.1a).

An intraperitoneal injection of LPS (5 mg kg⁻¹) resulted in increased plasma TNF-α levels and kidney damage (Fig. 1). Kidney damage was evaluated using plasma blood urea nitrogen (BUN), and mRNA levels of neutrophil gelatinase-associated lipocalin (*Ngal*) and kidney injury molecule-1 (*Kim-1*) as markers for early kidney injury and tubular damage and the number of apoptotic cells in renal tissue. Periodic acid-Schiff (PAS) staining demonstrated slight flattening of the brush border (Supplementary Fig. 1). Administering a single-dose of GTS-21 (20 mg kg⁻¹) 15 min before LPS administration decreased plasma BUN levels (Fig. 1c), *Ngal* and *Kim-1* expression (Fig. 1d), apoptotic cells (Fig. 1e, f) and plasma TNF-α levels (Fig. 1b). No histological changes were observed following LPS administration with or without GTS-21 administration following LPS injection (Supplementary Fig. 1).

**Both anti-inflammatory and renoprotective effects of GTS-21 are lost in macrophage-specific α7nAChR knockout mice.** To investigate the impact of α7nAChR on macrophages in vivo, we generated macrophage-specific α7nAChR KO mice by cross-breeding LysM-Cre and α7flox/flox mice (Supplementary Fig. 2). We then administered GTS-21 to mice with LPS-induced septic AKI, similar to WT mice (Fig. 2). The renoprotective effects were examined to determine whether they could be observed in macrophage-specific α7nAChR KO mice. The results showed that GTS-21 treatment reduced plasma TNF-α levels in littermate WT mice but not in macrophage-specific α7nAChR KO mice (Fig. 2b). Furthermore, littermate WT mice exhibited decreased *Ngal* gene levels and apoptotic cells, while macrophage-specific α7nAChR KO mice did not (Fig. 2d–f). There were no significant differences in plasma BUN, *Kim-1* gene expression, and histological changes (Supplementary Fig. 4) including among littermate WT mice.

**Single-cell RNA-sequencing of splenic cells reveals cell profiling affected by GTS-21 administration.** To comprehensively investigate the effect of GTS-21 on different cell types, we performed single-cell RNA-sequencing (scRNA-seq) of the whole spleen (Fig. 3). Spleens were harvested from LPS-induced septic WT mice and treated with GTS-21 under the same conditions as in the experiment in Fig. 1. We annotated each cell type using

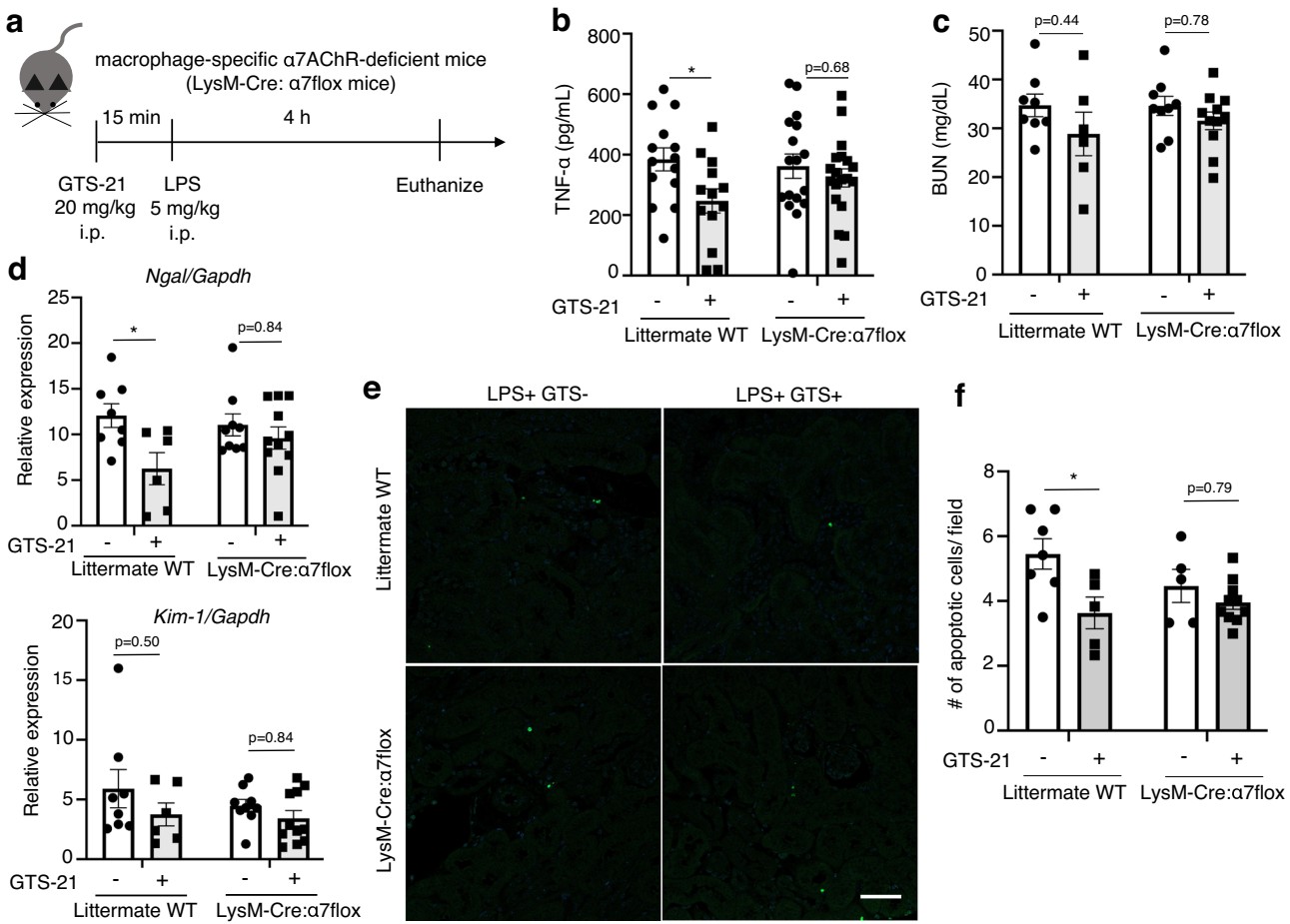

**Fig. 2 GTS-21 did not dampen the kidney injury in macrophage-specific α7AChR-deficient mice. a** Experimental design. Time-course and drug-dosage used are the same as in Fig. 1. After pre-treatment with GTS-21 or saline for 15 min, all groups of mice were intraperitoneally injected with LPS. Four hours later, the mice were euthanized, and blood and kidney samples were collected. **b** The decrease in TNF-α observed with GTS-21 administration in WT mice ($n = 13$–19 in each group). **c** Plasma BUN ($n = 6$–11 in each group). **d** Protection against LPS-induced kidney injury by GTS-21 ($n = 6$–11 in each group). **e, f** Representative pictures of TUNEL staining and several apoptotic cells. Scale bar = 50 μm. *$P < 0.05$, **$P < 0.01$, ***$P < 0.001$, ****$P < 0.0001$ (two-way ANOVA followed by Tukey's post hoc test). All data are presented as mean ± SEM. WT wild-type, GTS-21 3-(2,4-Dimethoxybenzylidene)- anabaseine dihydrochloride, LPS lipopolysaccharide, PAS periodic acid-Schiff, TNF-α tumor necrosis factor α, BUN blood urea nitrogen, qPCR quantitative polymerase chain reaction, TUNEL Terminal deoxynucleotidyl transferase dUTP nick end labeling, ANOVA analysis of variance, SEM standard error of the mean.

gene expression data of immune cell types from the Immunological Genome Project[57]. We integrated the four treatment groups and generated a uniform manifold approximation and projection (UMAP)[58](Fig. 3b). The annotated cell types are shown in Fig. 3c. GTS-21 administration did not significantly alter the population of immune cells in the spleen. However, since acetylcholine receptors on macrophages are important for exerting anti-inflammatory and renoprotective effects through CAP, therefore, we focused on the percentage of macrophages present. The proportion of macrophages did not change after GTS-21 treatment (Veh-Veh, 0.30%; Veh-GTS, 0.37%; LPS-Veh, 0.39%; LPS-GTS, 0.36%), as shown outside the pie chart in Fig. 3d.

**Ligand-receptor analysis reveals cell-cell interactions between macrophages induced by GTS-21.** Despite the anti-inflammatory effect exerted by α7nAChR on macrophages, there was no change in the proportion of macrophages. In our previous experiments, we demonstrated that the adoptive transfer of a small number ($1 \times 10^5$) of VNS-treated macrophages exerts strong anti-inflammatory and renoprotective effects[15, 16]. Therefore, we conducted a ligand-receptor (LR) analysis to evaluate cell-cell interactions in splenocytes (Fig. 4). The increase in plasma TNF-α

levels caused by LPS was significantly suppressed by GTS-21 when scRNA-seq was performed (Fig. 4a). Heatmaps in each group were assigned and compared between groups, focusing on ligand-receptor (LR) pairs that increased with GTS-21 administration during LPS-induced inflammation. LR pairs were increased by LPS stimulation compared to the vehicle (Veh) control (Fig. 4b, lower two maps vs. upper two maps). Comparing LPS-Veh and LPS-GTS, there were strong LR pairs between macrophages, DC, B, natural killer T (NKT), and T cells (Fig. 4b, lower two maps). The ligand on macrophages and its association with receptors on other immune cells is shown in the dot plot (Fig. 4c). Conversely, LR pairs of receptors of macrophages and ligands on other immune cells are shown in Fig. 4d. LR pairs was relatively increased in LPS group and LPS-GTS group. Regarding the LR pairs between macrophages, we identified 43 specific LR pairs were identified (Fig. 4e). Among the four treatment groups, the LPS-GTS group showed increased expression of LR pairs (Fig. 4e).

**GTS-21 increases the expression of adhesion factor between macrophages in the spleen.** To further explore the differences between macrophages in the spleen from the LPS and LPS-GTS

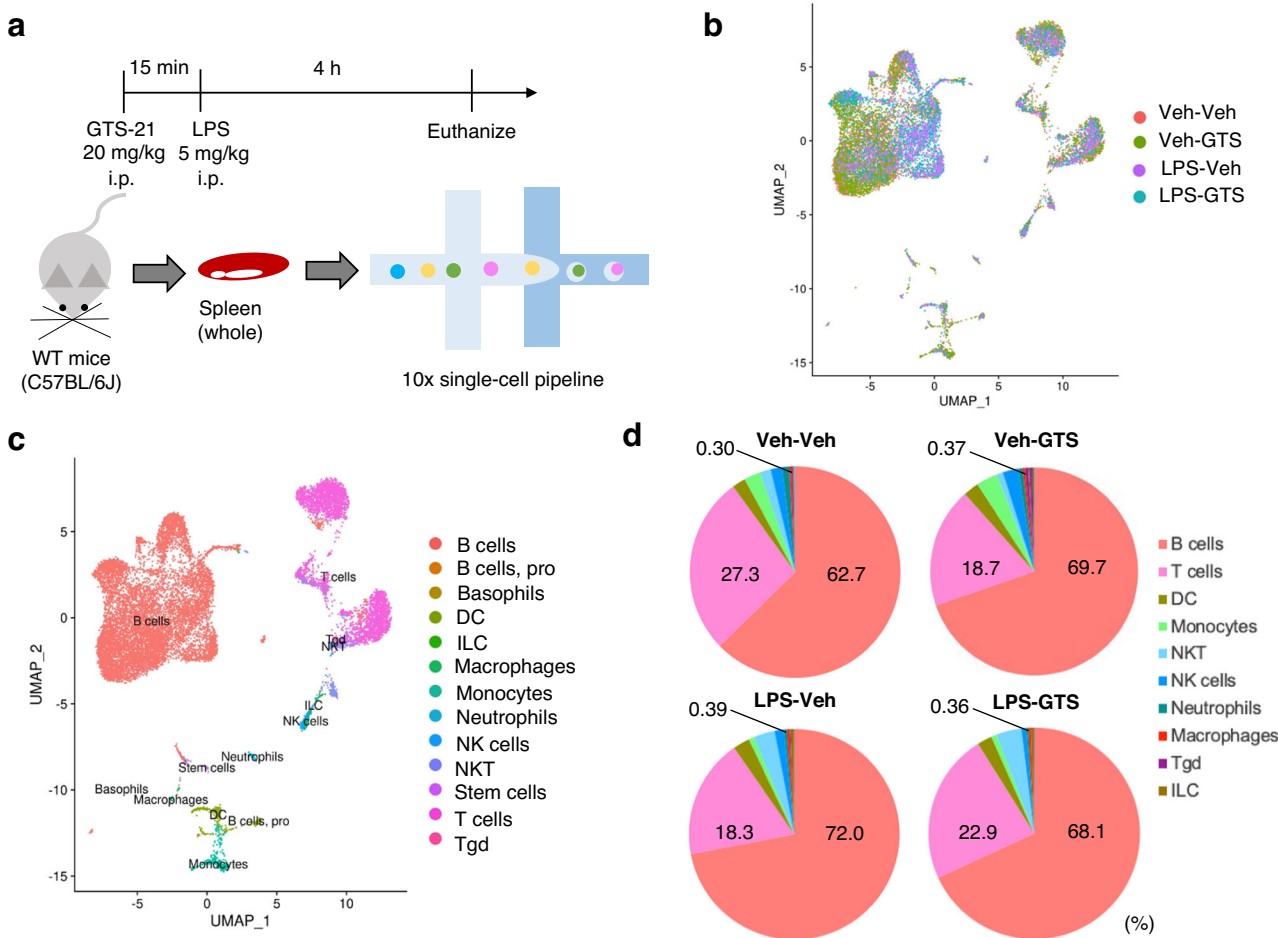

**Fig. 3 Single-cell RNA sequencing revealed that GTS-21 does not affect the proportion of immune cells in the spleen.** Single-cell RNA-seq was conducted using WT mice with or without LPS and GTS-21. **a** Schematic representation of the experimental procedure. Whole spleens removed from euthanized mice were isolated and subjected to single-cell RNA-seq. UMAP plots of the integrated single-cell data colored according to treatment group (**b**) and cell type (**c**). **d** The percentage of immune cells under each condition is shown in a pie chart. While total cell counts differed among groups (Veh-Veh, 4835 cells; Veh-GTS, 5616 cells, LPS-Veh, 3275 cells; and LPS-GTS, 2814 cells). WT wild-type, TNF-α tumor necrosis factor α, GTS-21 3-(2,4-Dimethoxybenzylidene)- anabaseine dihydrochloride, LPS lipopolysaccharide, UMAP uniform manifold approximation and projection plot, Veh vehicle control.

groups, we isolated macrophage-enriched samples from the spleen and conducted scRNA-seq (Fig. 5). WT mice treated with LPS and/or GTS-21 were euthanized after 4 h and their spleens were removed. Since the macrophage-enriched samples extracted by Magnetic cell sorting (MACS) contained B and T cells (Fig. 5a), we excluded them from analysis and further classified only macrophages into three clusters (Fig. 5b). Enrichment analysis of differentially expressed genes (DEGs) upregulated by GTS-21 revealed top gene ontology (GO)s associated with focal adhesion and cell-substrate junctions (Fig. 5c). Volcano plot also showed that integrin (*Itgal*) and chemokine (*Cxcr4*) were induced by GTS-21(Fig. 5d, e). To further investigate the interactions between macrophages, NicheNet was used to predict the ligand-receptor pairs influenced by GTS-21 administration (Fig. 5f–h). The top 16 predicted ligands produced by "sender" cells are shown in Fig. 5h. Among them, *Cd 80* and *Cd 86*, markers of pro-inflammatory macrophages, were also found, and their expression was predicted to be decreased by GTS-21 treatment. Circle plot shows the prediction that the ligands of the "sender" cells will influence the expression of the target genes observed in the "receiver" cells (Fig. 5f). The ligand of the three macrophage clusters (Mφ_1~3; Fig. 5b) predicted the impact on the largest cluster (Mφ_1; Fig. 5b). Ligands such as *Apoe, Itga4, Csf3, Pdgfb,*

and *Edn1* might be involved in regulating the expression of *Itgal* and *Ncf1* in cluster Mφ_1(Fig. 5b).

**GTS-21 increases macrophage-macrophage contacts, resulting in TNF-α reduction in vitro.** Given that GTS-21 might increase the LR pair expression between macrophage-macrophage inter-actions and promote the expression of adhesion molecules, including *Itgal*, we performed a transwell assay to confirm whether GTS-21 has a similar effect on macrophages in vitro. The number of migrating cells in RAW 264 cells, a murine macro-phage cell line, was determined by culturing them in the upper and lower sections separated by an insert (Fig. 6). RAW 264 cells in the upper chamber were labeled with CFSE to show migrated cells and cultured in medium with or without LPS/ GTS-21 for 4 or 24 h (Fig. 6a). After removing the insert, CFSE-labeled migrated cells were counted (following a 4 h waiting period); the number of migrated cells was similar in the four groups (Fig. 6b, c). When the observation time was extended to 24 h, the number of migrated cells increased significantly in the GTS-21 group (Fig. 6d, e), leading to an increase in macrophage-to-macrophage contact. The contact between macrophages was paradoxically reduced in the presence of α-bungarotoxin (α-Batx), an antagonist of α7nAChR (Supplementary Fig. 6).

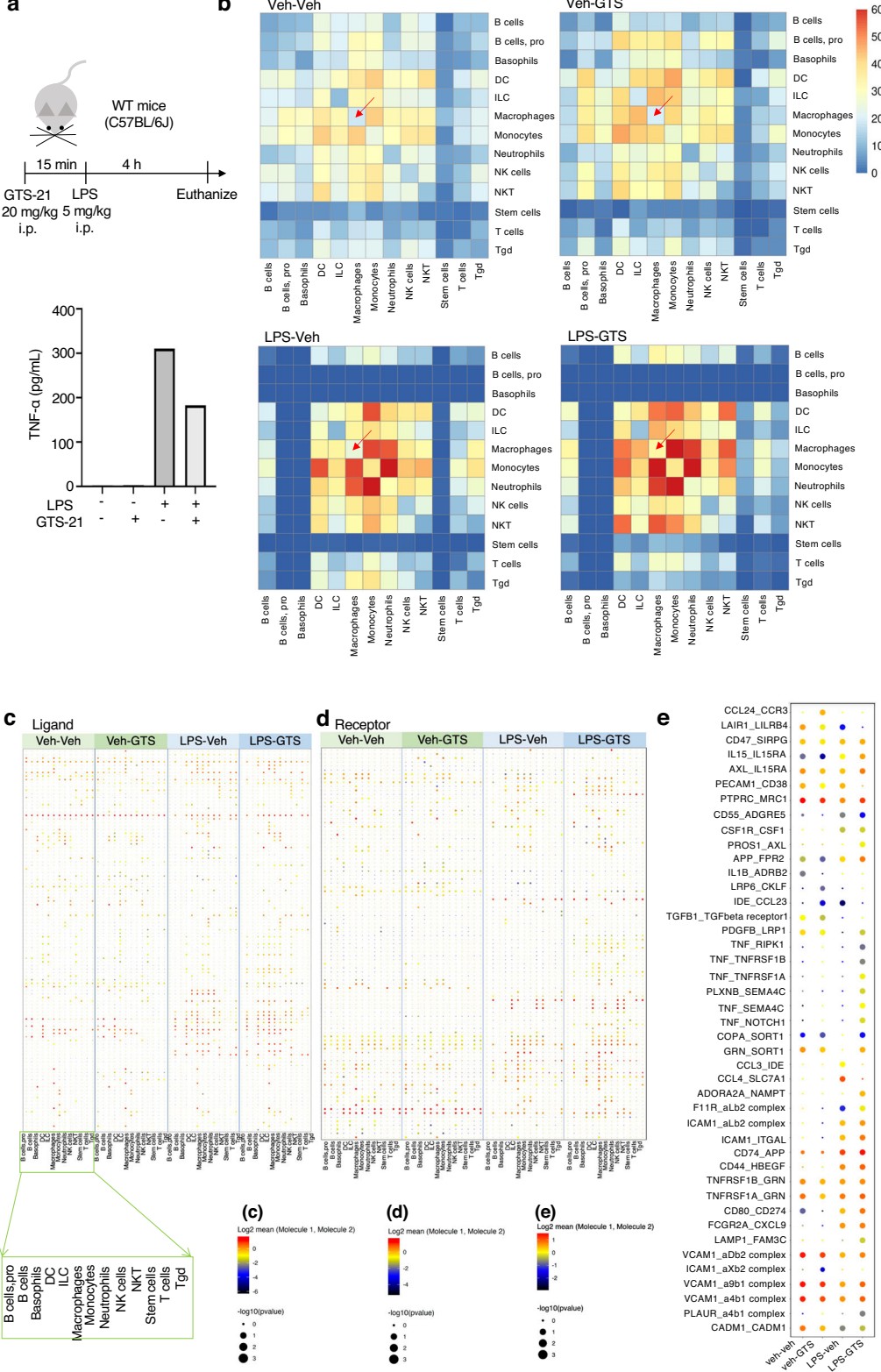

**Fig. 4 Ligand-receptor analysis revealed that GTS-21 increases macrophage-macrophage interactions. a** Plasma TNF-α level in mice subjected to single-cell RNA-seq. **b** Heatmap of the predicted count of ligand-receptor interactions between various immune cells in the spleen using CellPhoneDB. **c**, **d** Dot plot of the predicted ligand-receptor (LR) interactions from the macrophage side. LR pairs are shown in (**c**) when macrophages are ligands, whereas in (**d**) when macrophages are receptors. **e** Dot plot of the predicted ligand-receptor interactions focusing on the macrophage. GTS-21 3-(2,4-Dimethoxybenzylidene)-anabaseine dihydrochloride, LPS lipopolysaccharide, Veh vehicle control.

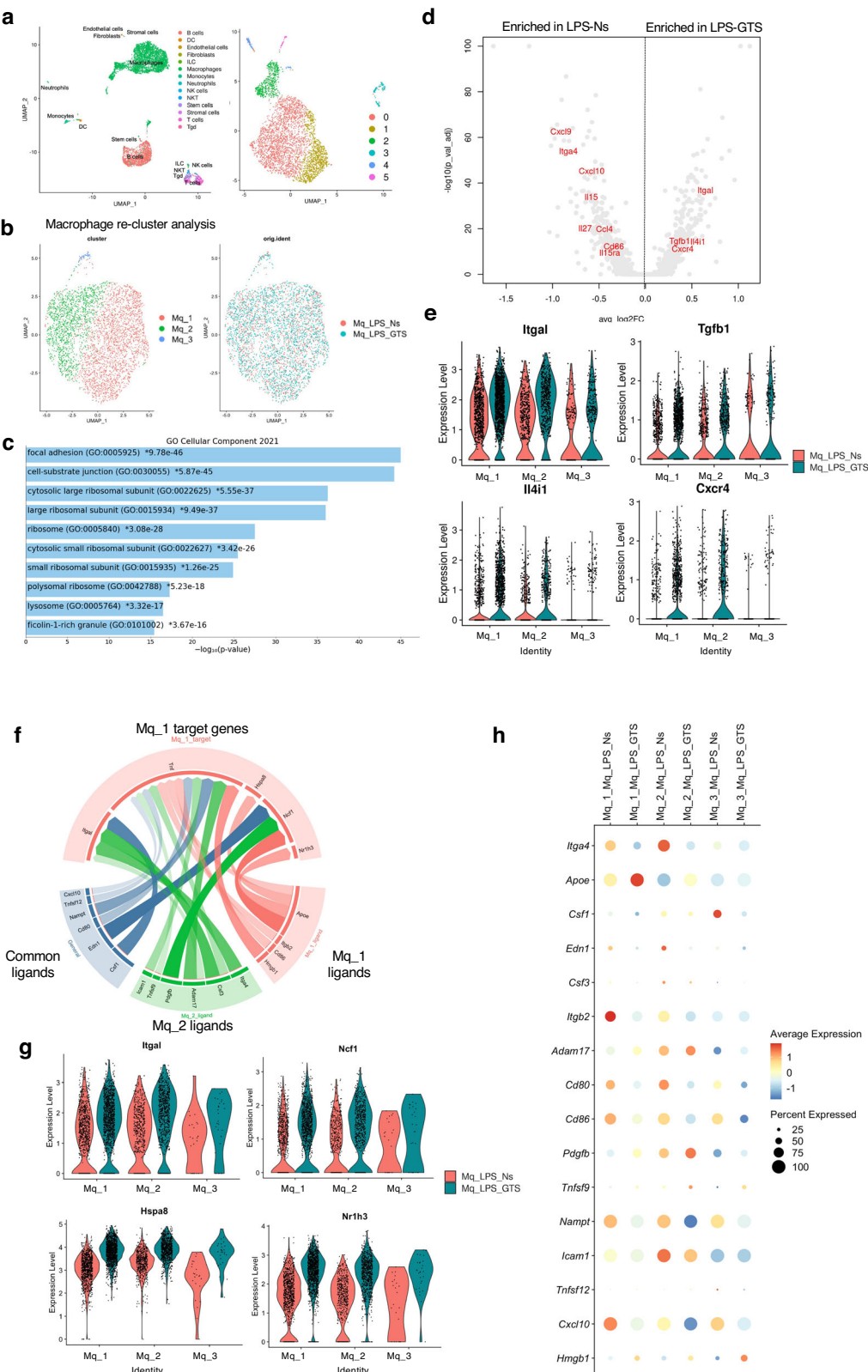

Furthermore, to investigate whether increased macrophage contact brought about anti-inflammatory effects, we performed the experiment shown in Fig. 6f using mouse and human macrophages. Fortunately, there was no cross-intersectionality between human and mouse TNF measurements using our usual ELISA. Therefore, we evaluated whether increasing human and mouse macrophage contact in co-culture would suppress inflammation.

U937 cells, a human monocyte cell line, were differentiated into macrophages using phorbol 12-myristate 13-acetate (PMA) and used as human macrophages (Fig. 6f). When RAW264 cells were stimulated with LPS (100 ng mL$^{-1}$), the murine TNF-α levels released by RAW264 cells in the medium were elevated, and TNF-α was suppressed by GTS-21 (100 μM). Administration of GTS-21 resulted in a decrease in the production of inflammatory

**Fig. 5 GTS-21 increased predicted ligand-receptor pairs in splenic macrophages.** WT mice received LPS-Normal saline (Ns) or LPS-GTS treatments for 4 h before being euthanized. Anti-F4/80 positive macrophages were collected from each spleen and scRNA-seq was performed. **a** UMAP projections of 6157 cells were integrated into each sample. Cell clusters included B cells, T cells, and other immune cells in addition to macrophages. **b** UMAP projection of re-clustered cells with only macrophages. The macrophages were further divided into three clusters (Mφ−1～3). **c** GO Enrichment analysis of the top genes in GTS-21 treatments. **d** Volcano plot of cytokine or chemokines induced enriched in LPS-Ns or LPS-GTS. **e** Violin plots show the gene expression levels increased by GTS-21 treatments in three macrophage clusters. **f** Circle plot depicting links between ligands and their predicted target genes for macrophage cluster1 and each cluster. Link color indicates the sender population of ligands, and link width and clarity correlate to the strength of the ligand-receptor integration and the activity score of the ligand, respectively. **g** Violin plot of target genes for Mφ_1 cluster. **h** Dot plot showing top 16 predicted ligands for each cluster. WT wild-type, GTS-21 3-(2,4-Dimethoxybenzylidene)- anabaseine dihydrochloride, LPS lipopolysaccharide, Ns normal saline, scRNA-seq single-cell RNA sequencing, UMAP plot Uniform Manifold Approximation and Projection plot, GO enrichment analysis, Gene ontology enrichment analysis.

---

cytokines such as *IL-6* and *IL-1β* from RAW 264 cells (Supplementary Fig. 8), and a tendency for macrophage phenotype to shift from pro-inflammatory to anti-inflammatory (Supplementary Fig. 9). The addition of human macrophages inhibited TNF-α production in RAW264 cells, as well as following GTS-21 treatment (Fig. 6g). Similar TNF-α reduction was observed when RAW264 cells were added instead of U937 (Supplementary Fig. 7). These findings indicate that the increase in macrophage-macrophage contact itself alters the phenotype of macrophages to an anti-inflammatory one.

**Splenectomy abolishes anti-inflammatory effects and kidney protection of GTS-21.** Next, we confirmed whether cell-cell interactions between macrophages in the spleen, which have been confirmed in in vitro experiments, are also essential for anti-inflammatory and kidney-protective effects in vivo. Since the spleen is the center of activity of the mononuclear phagocyte system, we removed the spleen to determine what happens when the contact between macrophages is reduced. As depicted in Fig. 7a, we conducted splenectomy before the administration of GTS-21 and LPS,and compared the results between the splenectomized mice and the sham-operated mice. The results showed that the administration of GTS-21 did not decrease the LPS-induced elevation of TNF-α (Fig. 7b) or the expression of renal injury markers, *Ngal* and *Kim-1* genes (Fig. 7c), in the splenectomized mice. In contrast, mice depleted of CD4 + T cells, a potential source of acetylcholine in the spleen, showed a decrease in TNF-α with GTS-21 administration (Fig. 7d). These results suggest that macrophages that receive α7nAChR signaling may be more important for CAP activation than acetylcholine transmission via splenic nerves and CD4 + T cells.

**Discussion**
Our study highlights the significance of macrophages and α7nAChRs in exerting anti-inflammatory and renal protective effects. Communication occurs between macrophages that receive cholinergic stimulation and those that do not in the spleen. The importance of this communication in exerting anti-inflammatory and renal protective effects has been demonstrated through in vitro and in vivo experiments involving splenectomy and CD4+ T cells depletion.

Though VNS exerts potent and reliable anti-inflammatory and renoprotective effects, we considered the possibility that it might affect other immune cells in addition to macrophages in the spleen. Although α7nAChR is required for CAP activation in macrophages and the spleen, the receiver of acetylcholine in vivo has not been clearly demonstrated. Recently, Guyot et al. found that one branch of the splenic nerve is cholinergic, not adrenergic, using macrophage-specific α7nAChR KO mice[59], however, a specific role for α7nAChR in splenic macrophages using cell-specific α7nAChR-deficient mice has not been demonstrated.

Notably, anti-inflammatory effects on other target organs, excluding those involving the spleen, have also been demonstrated in organs with direct vagal innervation of the gut and lungs, such as inflammatory bowel disease and acute lung injury[13,60–63]. For example, in a model of sepsis-induced acute lung injury, α7nAChR positive alveolar macrophages and neutrophils were present in bronchoalveolar lavage and nicotine stimulation ameliorated lung injury[64]. Thus, these organs, which are in direct contact with the external environment, are believed to possess distinct immune mechanisms and respond to VNS in addition to splenic CAP. In contrast, since parasympathetic connections have not been identified in the kidneys, splenic CAP is assumed to be important for exerting renoprotective effects in kidney injury. Furthermore, our previous studies using splenectomy and adoptive transfer of splenocytes suggest that the spleen plays an essential role in renal protection[14]. To further investigate the specific role of α7nAChR in macrophages, we generated macrophage-specific α7nAChR KO mice. As we hypothesized, these α7nAChR KO mice did not exhibit the GTS-21-induced decrease in TNF-α and kidney *Ngal* expression (Fig. 2), which was observed in WT mice induced by LPS (Fig. 1). This result suggests that α7nAChR signaling in macrophages has not only systemic anti-inflammatory effects, but also organ-protective effects.

Initially, we hypothesized that cholinergic signals might also be received by splenic cells other than macrophages, and that these cells might interact with other immune cells, given that various immune cells in the spleen express α7nAChR. To investigate this hypothesis, we used scRNA-seq to comprehensively study the splenocytes that receive cholinergic signaling. However, contrary to our expectations, GTS-21 did not cause any significant cellular remodeling (Fig. 3d). We also searched for genes that were altered by GTS-21 treatment in macrophages and other immune cells. However, changes in the expression of genes involved in the anti-inflammatory effect were not evident in any of the cells. Therefore, we considered the possibility that cells in the spleen interact with each other, resulting in anti-inflammatory effects. We performed LR analysis to detect cell-cell communication (Fig. 4), and LR analysis revealed that GTS-21 administration increased the interactions between macrophages and other immune cells, especially macrophages, DC, and NKT cells, under inflammatory conditions (Fig. 4b–d). Although macrophages and DC are known as phagocytes, were expected to have increased contact with T cells, the LR analysis showed an increase in macrophage-to-macrophage pairs where LR pairs were increased (Fig. 4e). Here, we focused on macrophage-macrophage interactions, considering that macrophage ligand expression levels must be subject to GTS-21-induced changes for α7nAChR-signaling macrophages to interact and that macrophages are the main source of TNF-α. Of the 43 LR pairs identified, the ligands are the TNF family ligands, chemokines (such as *CCL24*), and adhesion molecules (such as *ICAM-1* and *VCAM-1*). Furthermore,

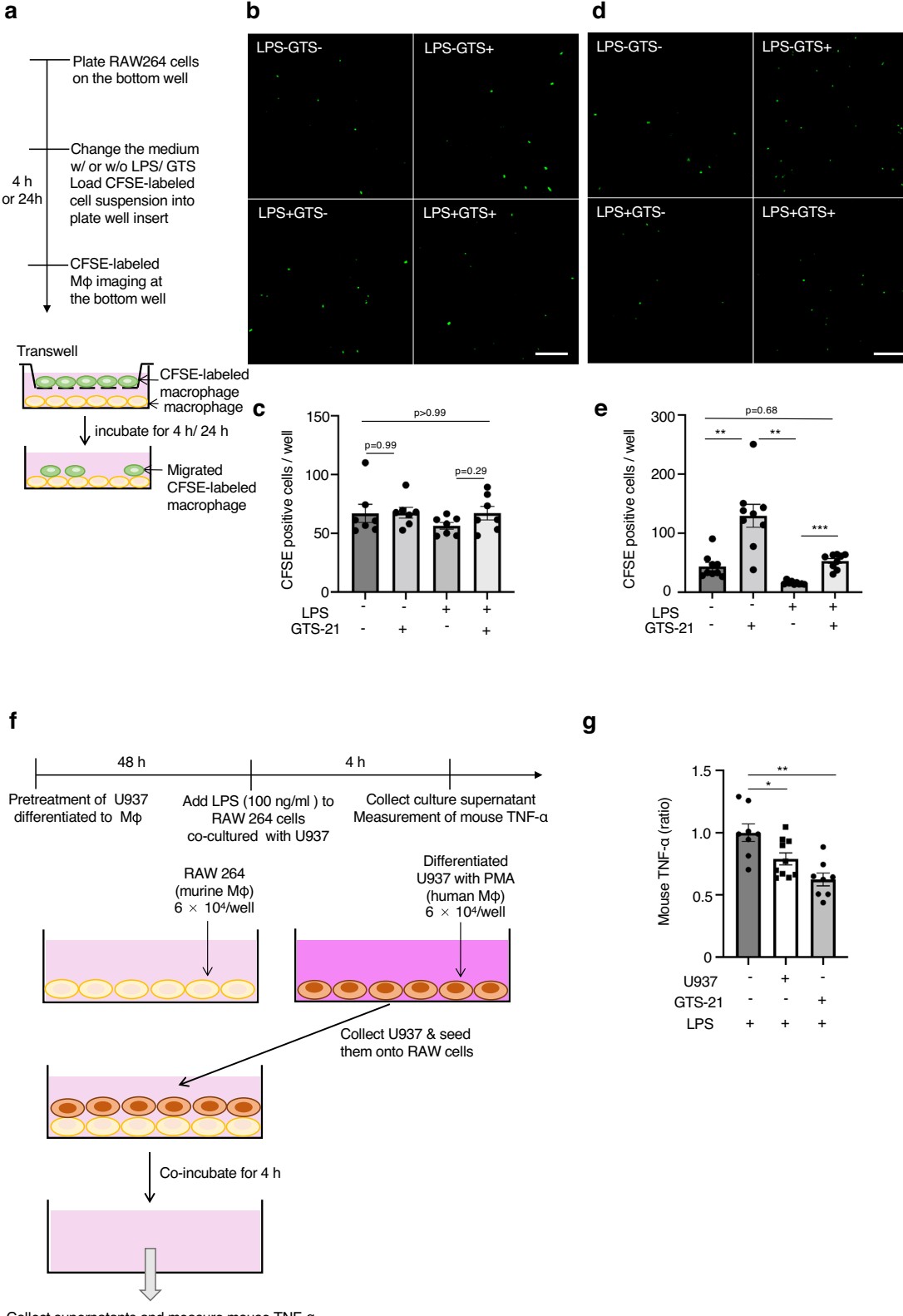

predictions of ligand-receptor communication affected by GTS-21 by NicheNet in macrophage-enriched samples (Fig. 5f–h) predicted that macrophages receiving GTS-21 expressed *Itga4* (Integrin Subunit Alpha 4) and *Csf3* (colony stimulating factor 3) ligands and predicted to induce expression of target genes such as *Itgal* (Integrin Alpha-L) and *Nr1h3* (genes involved in the regulation of macrophage function) in other cluster macrophages

(Fig. 5f). Indeed, *Itgal* was included in the DEGs upregulated by GTS-21 (Fig. 5d, e). We also found that GTS-21 might weaken the phenotype of macrophages that were changed proinflammatory (*CD80*, *CD86*) by LPS (Fig. 5h). This connection between macrophages was not limited to in vivo experiments; cell migration experiments using RAW 264 cells also showed that GTS-21 administration increases the number of cells that migrate

**Fig. 6 GTS-21 increased macrophage contacts using the transwell migration assay. a** The schema of this experiment. Migration assay of CFSE-labeled RAW 264 cells using a transwell system in different conditions. After 4 h or 24 h of incubation with and without LPS and GTS-21, the bottom wells were photographed. **b–e** Representative images of migrated cells are shown in (**b**; 4 h) and (**d**; 24 h). Bar graphs show the quantitative data of migrated cells (**c**; 4 h) (**e**; 24 h). **f, g** Co-culture of two types of macrophages suppressed the LPS-induced TNF-α induction. **f** Experimental protocols: Macrophages differentiated from the human monocyte cell line, U937, and mouse macrophage cell line, RAW 264, were co-cultured. The cells were treated with LPS and mouse TNF-α levels were measured 4 h later. As a control, RAW264 cells were treated with GTS-21 and LPS simultaneously. **g** The level of murine TNF-α. LPS-induced increase in TNF-α is also reduced when co-incubated with U937 cells. *P < 0.05, **P < 0.01, ***P < 0.001, (one-way ANOVA followed by Tukey's post hoc test). All data is presented as mean ± SEM. GTS-21 3-(2,4-Dimethoxybenzylidene)- anabaseine dihydrochloride, LPS lipopolysaccharide, CFSE carboxyfluorescein succinimidyl ester, TNF-α tumor necrosis factor α, PMA phorbol 12-myristate 13-acetate, ANOVA analysis of variance, SEM standard error of the mean.

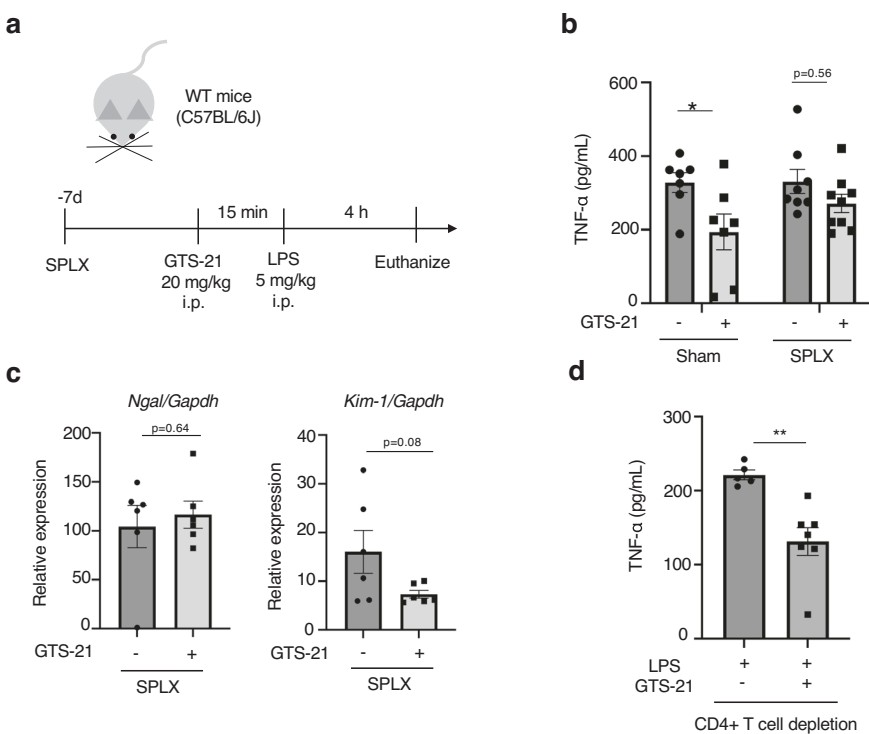

**Fig. 7 Splenectomy abolished the GTS-21 mediated anti-inflammatory and kidney protective effects in WT mice. a** Experimental design. Seven days prior to GTS-21 and LPS administration, the WT mice underwent splenectomy or sham operation (*n* = 6–9 in each group). **b** Plasma TNF-α levels and anti-inflammatory effect of GTS-21. **c** *Ngal* and *Kim-1* expression with following GTS-21 treatment. **d** Plasma TNF-α levels in CD4 + T cell-depleted mice with or without GTS-21. Plasma TNF-α was decreased in the GTS-21-treated mice (*P* = 0.003). *P < 0.05, (two-way ANOVA followed by Tukey's post hoc test (**b**), unpaired *t* test (**c**, **d**)). WT wild-type, GTS-21 3-(2,4-Dimethoxybenzylidene)- anabaseine dihydrochloride, LPS lipopolysaccharide, TNF-α tumor necrosis factor α, SPLX splenectomy, ANOVA analysis of variance.

during LPS-induced inflammation, resulting in the reduction of TNF-α production (Fig. 6f, g, Supplementary Fig. 7). These results suggest that macrophage-macrophage interactions play a major role in the anti-inflammatory effects of CAP activation.

It is important to note that the cell-cell interaction obtained here is the signaling of ligand receptors between single cells and does not include spatial information between cells. The lack of spatial information makes it difficult to identify the macrophages that influence this interaction. The spleen has two major functions: a filter function that traps pathogens or antigens and aged or abnormal red blood cells in the blood, and an immune function that regulates innate and adaptive immunity. It is anatomically divided into red pulp, white pulp, and the marginal zone that separates them[65]. Tissue-resident macrophages in the spleen are classified into red pulp macrophages, white pulp macrophages, tingible macrophages (located in the germinal center), and marginal zone macrophages; each with different morphologies and functions[66]. Furthermore, it is assumed that circulating macrophages in the spleen increase under inflammation-induced

conditions. However, we did not identify the type of macrophage used found in our study. It would be challenging to determine which macrophages in the spleen receive cholinergic signals and influence each other. The use of spatial transcriptome analysis, which has rapidly improved in recent years[67], will allow for a more detailed analysis of macrophage localization.

Our previous adoptive transfer experiment of VNS-treated splenocytes confirmed the renoprotective effect of a small number of macrophages ($1 \times 10^5$)[15,16]. Although most of TNF-α is produced by macrophages, the population of macrophages accounts for less than 1% of the splenocytes (Fig. 3d). There is no evidence to explain how such a small percentage of macrophages have a strong ability to ameliorate systemic inflammation and renal damage. Surprisingly, in the macrophage-macrophage co-culture experiment, we found that macrophage contact itself suppressed TNF-α production (Fig. 6f, g and Supplementary Fig. 7). When this macrophage-macrophage interaction was deficient during splenectomy, the anti-inflammatory effect was abolished, as expected (Fig. 7a–c). These observations show that splenic CAP

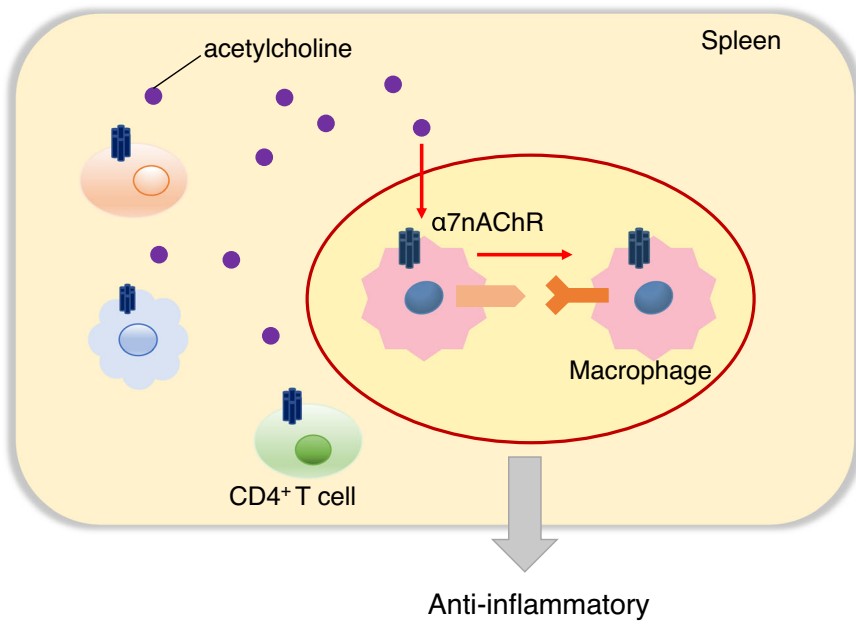

**Fig. 8 Scheme of our new findings of the CAP.** In the spleen, among the many types of immune cells that express α7nAChRs, α7nAChR on macrophages receive acetylcholine. Macrophages that receive cholinergic signals trigger cell-cell interactions between macrophages. This response results in the suppression of systemic TNF-α production and the reduction of kidney damage. CAP cholinergic anti-inflammatory pathway, TNF-α tumor necrosis factor α.

activation has an anti-inflammatory effect by activating α7nAChR on macrophages and further inducing macrophage-macrophage interactions in the spleen. Interestingly, these signal inductions can be exerted only under inflammatory conditions.

Our study had certain limitations. First, scRNA-seq allowed us to find interactions between macrophages and exert anti-inflammatory effects, and some candidate genes related to cell adhesion or functional regulation and phenotypic change of macrophages were detected (Figs. 4 and 5). However, it has not been confirmed whether these altered genes can interact with macrophages in vivo. Next, in experiments using macrophage-specific knockout mice (Fig. 2), it was difficult to find differences without increasing the number of mice. The large variability may be due in part to the use of a wide range of age groups and sexes of mice because elderly mice are more susceptible to LPS than young mice[68]. However, even when the sex and age of mice were precisely matched, the variability remained large (Supplementary Fig. 3). It might also suggest that the activation of the splenic CAP through α7nAChR in macrophages occurs only in the very early stages of injury. The results show that TNF-α and *Kim-1* were decreased in macrophage-specific α7nAChR KO mice 24 h after LPS administration (Supplementary Fig. 5), which may support the above hypothesis. In addition, it should be noted that GTS-21 used in this study is an agonist not only for the α7 nicotinic receptor but also for the α4β2 subtype and has been reported to act as a silent agonist[69,70]. Recently, it has been reported that the α9 subunit nicotinic acetylcholine receptor is involved in the activation of CAP[71,72], and the verification of the role of α9 may be necessary for the future. Finally, we previously reported that VNS induces *Hes1* genes via α7nAChR in peritoneal macrophages, resulting in organ-protective effects[15]. It is thought that VNS will probably exert renal protection in combination with a non-splenic pathway. The results obtained in this study might need validation using different mice models.

In summary, this work established that activation of the CAP occurs via the α7nAChR of macrophages in the spleen, and the signal enhances macrophage-macrophage interactions (Fig. 8).

Macrophage-macrophage signaling leads to anti-inflammatory effects that are exerted only under inflammation-induced conditions. Since splenic CAP is involved in inflammatory diseases other than kidney injury, this new finding will contribute to other inflammatory diseases.

## Material and methods

**Animal experiments**. C57BL/6 J male mice (8–12 weeks old, 20–25 g) purchased from CLEA (Tokyo, Japan) were used as WT mice. To generate macrophage-specific α7nAChR KO mice, lysozyme M (*LysM*)-Cre and α7nAChR flox/flox (B6(Cg)-*Chrna7tm1.1Ehs*/YakelJ, Jackson Laboratory, Bar Harbor, ME, USA) mice were crossbred for several generations. LysM-Cre: α7 flox mice were genotyped using tail PCR based on the protocol provided by the Jackson Laboratory using a MightyAmp Genotyping kit (Takara Bio Inc., Shiga, Japan). The primer sequence used in tail-PCR is listed in Supplementary Table 1. LysM-Cre: α7 flox male and female mice (8–18 weeks old) were used for the experiments. General anesthesia (0.3 mg kg$^{-1}$ medetomidine, 5 mg kg$^{-1}$ butorphanol, and 4 mg kg$^{-1}$ midazolam) was administered for all surgeries and euthanasia.

**LPS-induced AKI model and GTS-21 treatment**. Septic AKI mouse models were created through intraperitoneal injection of lipopolysaccharides (LPS) from *Escherichia coli* O111:B4 and purified using phenol extraction (Sigma-Aldrich, Cat#L2630, LPS; 5 mg kg$^{-1}$). As in our previous study, GTS-21 (20 mg kg$^{-1}$, Sigma-Aldrich, Cat#SML0326) was injected intraperitoneally 15 min before LPS administration. The vehicle control received an equal volume of normal saline. Four hours after the LPS injection, the mice were euthanized.

**Splenectomy**. Splenectomy was performed under general anesthesia. A small incision was made on the back of the mouse, the splenic vessels were ligated, and the spleen was removed. The sham-operated mice underwent skin incisions only. Seven days

after the operation, the mice received an intraperitoneal injection of LPS, followed by a GTS-21 injection. Four hours later, the blood and kidney samples were collected.

**CD4 positive T cells depletion**. Two groups of WT mice received anti-CD4 (clone: GK 1.5, #BE0003-1, Bio X Cell, Lebanon, NH, USA) antibody or rat IgG2b isotype control (clone: LTF-2, #BE0090, Bio X Cell) treatments (200 μl per mouse, i.p. injection) three days before experiments. The efficacy of CD4 + T cell depletion was confirmed with spleen and whole blood cells by flow cytometry analysis (Attune NxT Flow Cytometer, Thermo Fisher Scientific) (Supplementary Fig. 10).

**Validation of CD4 positive T cells depletion**. To confirm CD4 positive T cell depletion using Flow cytometry, single cell suspensions were prepared from spleen and whole blood samples. Spleens were mushed using a plunger and passed through a 40 μm filter with 10 mL ice-cold PBS. The cell suspension were centrifuged at $500 \times g$ for 5 min, then the supernatant was discarded. Whole blood cells were subjected to red blood cells (RBC) removal using RBC lysis buffer (5 mL). After blocking nonspecific Fc binding with anti-mouse CD16/32 (2.4G2), the cell suspensions were incubated with the following antibodies: anti-mouse CD4-FITC (GK1.5, Thermo Fisher Scientific), CD8a-eFloro 450 (53-6.7, Thermo Fisher Scientific). 7-AAD (Thermo Fisher Scientific) was used to exclude dead cells. Flow cytometry was performed on Attune NxT (Thermo Fisher Scientific), and the data were analyzed using Flow Jo CE softwear.

**RNA extraction and real-time quantitative PCR**. Messenger RNA was extracted from a quarter of the left kidney using RNAiso Plus (Takara Bio Inc., Shiga, Japan). RNA was extracted from the cells using the FastGene RNA Premium Kit (NIPPON Genetics, Tokyo, Japan). Reverse transcription was performed using the PrimeScript RT Master Mix (Takara Bio Inc., Shiga, Japan). Synthesized cDNA was used as the template for quantitative real-time PCR, which was performed using iTaq Universal SYBR Green Supermix (Bio-Rad) on a CFX Connect Real-Time PCR Detection System (Bio-Rad). Glyceraldehyde 3-phosphate dehydrogenase (GAPDH) was used as the internal control. Furthermore, the relative gene expression levels were calculated using the comparative cycle threshold (CT; $2^{-\Delta\Delta Ct}$) method. The primer sequences are listed in Supplementary Table 2.

**Preparation of peritoneal macrophages**. Four days before the experiments, mice were received 3% Thioglycollate (#225710, Becton, Dickinson and Company, Franklin lakes, NJ) 2 mL i.p.. The day of experiments, the mice were euthanized, 10 mL of ice-cold sterile PBS was injected into peritoneal cavity. The injected fluid was collected after the peritoneum was gentry massaged. The collected fluid was centrifuged at $500 \times g$ for 5 min, then the supernatant was discarded. DNA was extracted from the resulting cell pellets using a DNeasy Blood & tissue kit (# 69504, Qiagen, Hilden, Germany).

**Measurement of TNF-α**. Levels of TNF-α in plasma and culture medium from cell supernatant were measured using ELISA with the TNF alpha Mouse Uncoated ELISA Kit with Plates (88-7324-22, Thermo Fisher Scientific, Waltham, MA), according to the manufacturer's instructions. The Synergy LX (BioTek Instruments) was used for the ELISA plate reader.

**Measurement of renal function and apoptosis in kidney tissue**. To evaluate renal function, the plasma BUN levels were measured by SRL, Inc. (Osaka, Japan). Terminal deoxynucleotidyl

transferase dUTP nick end labeling (TUNEL) staining was used to evaluate apoptotic cells in the kidney tissues using Click-iT™ Plus TUNEL Assay Kits for In Situ Apoptosis Detection (C10617, Thermo Fisher Scientific, Waltham, MA). One-quarter of the left kidney was fixed with 4% paraformaldehyde (PFA) and embedded in paraffin. Tissues were sliced into 4-μm sections and stained according to the manufacturer's protocol, and the slides were examined using a confocal microscope (LSM 800; Zeiss, Oberkochen, Germany).

**PAS staining**. One-quarter of the left kidney was fixed with 4% paraformaldehyde (PFA) and embedded in paraffin. Tissues were sliced into 3-μm sections and stained with periodic acid-Schiff (PAS) to evaluate the tubular damage. Tubular injury scores were evaluated based on the proportion of injured tubules as follows: 0. None; 1, <25%; 2, 25–50%; 3, 50–75%; and 4, >75%.

**Cell culture**. Murine macrophage cells, RAW 264 (RIKEN BRC CELL bank, Ibaraki, Japan), were incubated in Dulbecco's Modified Eagle's Medium (DMEM) high glucose (Sigma-Aldrich, St Louis, MO, USA) supplemented with 10% fetal bovine serum (FBS) (Sigma-Aldrich), 50 U mL$^{-1}$ penicillin, and 50 μg mL$^{-1}$ streptomycin. U937 cells (JCRB Cell Bank, Osaka, Japan), a human monocyte cell line, were cultured in RPMI-1640 (Sigma-Aldrich, St Louis, MO, USA) supplemented with 10% FBS (Sigma-Aldrich). The cells were incubated at 37 °C in a humidified atmosphere containing 5% $CO_2$.

**Transwell migration assay**. Migration assays were performed using 24 well transwell plates with 8.0 μm pore size inserts. RAW264 cells were cultured in 24 well plates at a density of $4 \times 10^5$ cells per well. To identify the migrated cells, RAW264 cells on the upper side were labeled with green fluorescence using the CellTrace CFSE Cell Proliferation Kit (C32554; Thermo Fisher Scientific) according to the manufacturer's instructions. Carboxyfluorescein succinimidyl ester (CFSE)-labeled cells were seeded into the upper chamber. GTS-21 (100 μM) or α-bungarotoxin (#203980, Sigma-Aldrich) (1 μg ml$^{-1}$) was added to a medium containing LPS (100 ng ml$^{-1}$) or PBS as a control, and the number of migrated cells was evaluated after 4 or 24 h. The same treatment medium was then added to the upper and lower chambers. Migrated cells were photographed using a confocal microscopy (C2+ system; Nikon Corporation, Tokyo, Japan), and the images were visualized using IMARIS software (Bitplane, Zurich, Switzerland). The number of CFSE-labeled migrated cells were counted in every three random rows in the field, and the mean value for each group was calculated.

**Macrophage co-culture experiment**. To investigate the anti-inflammatory effects of macrophage-to-macrophage contact, human macrophages were added to mouse macrophages and co-cultured, and murine TNF-α levels in the medium were measured after administering LPS. Human macrophage-differentiated U937 cells were used. U937 cells were treated with phorbol 12-myristate 13-acetate (PMA; Sigma-Aldrich) for 48 h, which allowed the cells to differentiate into macrophages. After washing the cells once with PBS, they were seeded onto pre-seeded RAW264 cells. RAW 264 cells were pre-seeded in 24-well plates at $6 \times 10^4$ cells per well, and the same number of differentiated U937 cells were seeded and co-cultured. As a control, RAW264 cells were treated with GTS-21 (100 μM) for 15 min prior to LPS stimulation (100 ng mL$^{-1}$) for 4 h and the culture medium was collected.

**Single-cell RNA-seq and preparation of single cell suspension**. A cocktail of 20 U mL$^{-1}$ DNase (Promega Inc. WI, USA.

Cat#M6101) and $2\,mg\,mL^{-1}$ collagenase type 1 (Worthington Corp., NJ, USA, Cat#CLS1), $1\,mg\,mL^{-1}$ collagenase type 2 (Worthington, Cat#CLS2), and $1\,mg\,mL^{-1}$ dispase (Roche, Cat#4942078001) was dissolved in DMEM high glucose (Wako) containing 10% FBS, which was used as a digestion buffer. Whole spleens were harvested and strained using 40 μm filters (SPL Life Sciences Co.Ltd. Pocheon, Korea) with a digestion buffer to obtain single-cell suspensions. The suspensions were subsequently collected in 15 mL tubes with 2.5 mL digestion buffer, pipetted into wells, and incubated in a shaking water bath at 37 °C for 10 min. The supernatant was collected in ice-cold high-glucose DMEM containing 10% FBS to stop enzymatic reactions. Fresh digestion buffer (2.5 mL) was added to the remaining cell pellets and subsequently incubated in a shaking water bath at 37 °C for 10 min. This series of experiments was repeated four times. The single-cell suspensions obtained by collecting the supernatant were passed through a 70-μm cell strainer and further passed through a 40-μm cell strainer. After centrifugation at 300 g for 5 min at 4 °C, the supernatant was discarded. One milliliter of digestion buffer was added to the cell pellets, pipetted well, and diluted up to 10 mL; the solution was then passed through a 40-μm cell strainer thrice. Single-cell suspensions were placed on ice, and the cell numbers were counted. Cell viability and singlets were confirmed using flow cytometry (Attune NxT Flow Cytometer, Thermo Fisher Scientific, Inc., Waltham, MA, USA). This method generated a single-cell suspension with viability greater than 90%. To obtain macrophage-enriched scRNA-seq, macrophages were isolated from spleen single-cell suspension labeling with anti-F4/ 80 Microbeads (130–110–443; Miltenyi Biotec, Bergisch Gladbach, Germany) using the magnetic cell separation method.

**Single-cell RNA-seq barcoding and synthesizing cDNA libraries.** Seven thousand cells were loaded onto a 10x Genomics Chromium instrument (10x Genomics, Pleasanton, CA, USA) to create single-cell Gel Bead-in-Emulsions (GEMs). cDNA libraries were constructed using 10x Chromium Single cell 3' Reagent Kits v3.1, following the manufacturer's instructions. Briefly, amplified cDNA products were cleaned using the SPRI Select Reagent Kit (10x Genomics). Indexed sequencing libraries were constructed and barcoded sequencing libraries were quantified using the Qubit 2.0 ds HS Assay Kit (Invitrogen). The quality of the libraries was checked using an Agilent 2200 Tapestation System (Agilent Technologies,Inc. Santa Clara, CA). Finally, libraries were sequenced using the Illumina NovaSeq platform.

**Single-cell RNA-seq data analysis.** The fastq files of four treatment groups (veh-veh, veh-GTS, LPS-veh, and LPS-GTS) were processed using the Cell Ranger v.3.1.0 (10× Genomics) count pipeline against the mm10 (v.3.0.0) mouse reference sequence. We used Seurat v.4.0.6[73] for detailed analysis. Cells with greater than 5% mitochondrial RNA or < 1,000, or > 4,000 detected genes were excluded. The data was subsequently normalized using the "NormalizeData" function, searched for 5,000 highly variable genes using the "FindVariableFeatures" function, and integrated four treatment groups using the "FindIntegrationAnchors" and "IntegrateData" functions. We visualized the integrated data using UMAP[58] colored by group identities (Fig. 4b). We also used SingleR v.1.4.1[57] against the Immunological Genome Project dataset ("ImmGenData")[74] for cell-type recognition. The estimated cell types were visualized using UMAP (Fig. 3c and Fig. 5a, b). Cell–cell communication analysis between cell types was performed using CellPhoneDB[75] for each treatment group. The mouse genes were converted into the human genes using the biomaRt package[76], and "cellphonedb method statistical_analysis" was run with using the

"—threshold=0.2" parameter. The predicted ligand-receptor interactions between cell types in each treatment group were visualized using a heatmap (Fig. 4b). We also visualized the predicted ligand-receptor (LR) interactions related to macrophages using the "cellphonedb plot dot_plot" command (Fig. 4c–e). Macrophage ligand LR pairs and receptors on other immune cells are shown in Fig. 4c Macrophage receptor LR pairs and ligands on other immune cells are shown in Fig. 4d and LR pairs of macrophage ligands and macrophage receptors are shown in Fig. 4e.

Gene set enrichment analysis of top 1,000 genes induced by GTS-21 in Mφ_cluster 1, the largest cluster of macrophage, was performed by Enricher (https://maayanlab.cloud/Enrichr/) and visualized with Enrichr Appyter (https://appyters.maayanlab.cloud/#/) (Fig. 5c).

Analysis of potential ligand-receptor interactions in macrophage-enriched samples was performed using NicheNet R package[77] according to the code deposited in GitHub (https://github.com/saeyslab/nichenetr), linking potential ligands expressed in sender cells to their target genes and their corresponding receptors that are differentially expressed in receiver cells. The ligand-receptor pairs were visualized in a code diagram using the R package circlize[78] (Fig. 5f–h).

**Statistical analysis.** Data are expressed as mean ± standard error of the mean (SEM). The Student's $t$ test was used for comparisons between the two groups. One-way, two-way analysis of variance (ANOVA) followed by Tukey's post-hoc test was used for three or more groups. Statistical significance was defined as $P < 0.05$. All analyses were performed using the GraphPad Prism 9 software (GraphPad Inc.).

**Study approval.** All animal experiments were conducted in accordance with the Guidelines for the Care and Use of Laboratory Animals, approved by the University of Tokyo Graduate School of Medicine and Nagasaki University (M-P18-051 for the University of Tokyo and 2006101636 for Nagasaki University).

**Reporting summary.** Further information on research design is available in the Nature Portfolio Reporting Summary linked to this article.

## Data availability
Raw sequencing data in this study have been deposited in the DDBJ Sequence Read Archive under the accession numbers DRA014634 and DRA016213. The sequenced data were deposited in the Genomic Expression Archive (GEA) under accession number E-GEAD-529, E-GEAD-608 and E-GEAD-609. All other data can be obtained from the corresponding author upon reasonable request.

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

## Acknowledgements

We are grateful to Dr. Ryoichi Mori (Department of Pathology, Nagasaki University Graduate School of Biomedical Sciences) for photographing the confocal microscopes. We also thank Ms. Minako Sakibe, Ms. Ryoko Yamamoto, Ms. Yoko Komori, Ms. Kurara Nakamura at the Department of Physiology of Visceral Function and Body Fluid, Nagasaki University Graduate School of Biomedical Sciences for their technical support. We would like to thank Editage (www.editage.com) for English language editing. This work was supported by Grant-in Aid for Research Activity start-up (JSPS KAKENHI grant 21K20874 (to Y.N.)); Grant-in Aid for Young Scientists (JSPS KAKENHI grant 22K16107 (to Wu CH) and 23K15251(to Y.N.)); Grant-in Aid for Research Activity start-up, Young Scientists and Scientific Research (B) (18H06192, 20K17242 and 22H03090), AMED PRIME (JP22gm6210013), JST FOREST (JPMJFR210J), MSD Life Science Foundation, Kidney Foundation (JKFB18-3), Salt Science Research Foundation (No.1919 and No. 23C5), Smoking Research Foundation, Yukiko Ishibashi Foundation, Mochida Memorial Foundation, Takeda Science Foundation, Astellas Foundation for Research on Metabolic Disorders, Suzuken Memorial Foundation, Tokyo Biochemical Research Foundation, Japan Kidney Association/Japan Boehringer Ingelheim Joint Research Project Grant, The Naito Foundation, Daiichi Sankyo Foundation of Life Science, The Uehara Memorial Foundation and Terumo Life Science Foundation to TI and Kyowa Kirin to RI.

## Author contributions

Y.N. performed the main experiments, analyzed the data, and wrote the original draft; H.M. and C.-H.W. analyzed the scRNA-seq data; D.F., R.U., Y.H., and T.I. performed or contributed to the animal experiments; M.K., S.Y., T.K., I.K., and S.N. aided in the scRNA-seq analysis; Y.W., M.N., R.I., and T.I. supervised this study and revised the manuscript. All authors approved the final version of the manuscript.

## Competing interests

The authors declare no competing interests.

## Additional information

**Peer review information** : *Communications Biology* thanks Bruno Bonaz and the other, anonymous, reviewer(s) for their contribution to the peer review of this work. Primary Handling Editors: Zhijuan Qiu and Karli Montague-Cardoso. A peer review file is available.

