## [Peer Review File · Communications Biology]

Reviewers' comments:

Reviewer #1 (Remarks to the Author):

The cholinergic anti-inflammatory pathway (CAP) has anti-inflammatory properties through vagal efferents (inflammatory reflex). Alpha 7 nicotinic acetylcholine receptor ($\alpha 7$ nAChR) positive macrophages have an important role in this pathway. Vagus nerve stimulation (VNS) improves acute kidney injury in rodent models. In this study, the authors attempted to elucidate the function of the $\alpha 7$ nAChR in macrophages in vivo using newly generated macrophage-specific $\alpha 7$ nAChR-deficient mice. Their results indicate that $\alpha 7$ nAChR signaling increases macrophage-macrophage interactions in the spleen and has kidney protective effects.

This is an original, interesting, and well done study with potential therapeutic implications by a group referent in the field. I have only minor comments:

- the cholinergic anti-inflammatory pathway through spleen macrophages is a "non-neuronal cholinergic pathway" by comparison to the one of the vagus nerve. Please specify.
- the authors did not explore the effect of sympathetic/vagal (splenic) denervation in their model.
- limitations of the study: the authors observed a large variability due to a wide range of age groups and sexes of mice (page 19, line 4). They had better to select these groups for their study.

Reviewer #2 (Remarks to the Author):

Nakamura and colleagues showed that $\alpha 7$ AChR positive macrophages are involved in the cholinergic anti-inflammatory pathway using LysM-Cre. $\alpha 7$ nAChR flox mice. They performed scRNAseq on the whole spleen to determine whether the $\alpha 7$ nAChR agonist GTS-21 induced marked changes in specific cell populations. Among others, macrophages increased their cell-cell interactions. To this end, the authors used mixed mouse and human cell line culture to show that GTS-21 increased cell-contact, causing TNF reduction.

Major

Figure 2: GTS-21 does not seem to be reproductively inducing anti-inflammatory effects, making it difficult to truly state that CAP is relayed by $\alpha 7$ nAChR+ macrophages: In contrast to the WT mice of Figure 1, the littermate WT of Figure 2 namely did not have reduced BUN and KIM levels following GTS-21 administration. Similar observations were made in Supplementary Figure 1. In addition, both in Figure 1 and 2 GTS-21 did not reduce tubular injury score. Thus, the authors are overstating that GTS-21 has renoprotective effects. Maybe better to show that GTS-21 reduces apoptosis following LPS as showed by Gao et al (2017).

3: Since there are so few macrophages sequenced, the authors should not state that GTS-21 does not induce changes in the macrophage transcriptome. In fact, they should repeat the scRNA seq and FACS-sort to enrich for macrophages. In addition, they do not show that these macrophages express $\alpha 7$ nAChR or other nicotinic receptors. I also miss some volcano plots showing differential expressed genes of macrophages and some SCENIC analysis to unravel to transcription factors leading to the macrophage phenotype in the spleen.

Figure 4: The authors ignore the fact that DC and NK cells have increased interactions with macrophages between LPS-Veh vs LPS-GTS. In addition, they do not make a deep characterization of macrophage phenotype following the different treatments. They only describe how many LR pairs are different, but do not discuss the genes until the discussion.

Figure 5: Even though the authors state the GTS-21 increased cell-cell contact based on the transwell experiments, it just appears that GTS-21 increases migration. What factor do the author think that is

inducing this transmigration?

Figure 6: Do the authors obtain similar results if they just double the concentration of RAW cells from 6×10^4 cells to 12×10^4 ?

Figure 7: The authors did splenectomy to determine if cell-cell interactions between macrophages in the spleen is essential for the CAP. It however seems that this experiment just proves the spleen is required for the CAP, not specifically the macrophages.

Minor

Instead of research papers, a lot of review papers are cited in the introduction

Instead of 9, 10: VNS in Crohn: 26920654, 32515156; colitis: 34307422 31133776

Instead of 58  23929694

Reviewer #3 (Remarks to the Author):

The purpose of this study is to clarify that CAP activation is achieved through $\alpha 7$ nAChR of macrophages in the spleen. And this signal enhances the interaction between macrophages and macrophages, so as to achieve anti-inflammatory and organ protection. The author has done a lot of work, and the conclusion is persuasive to some extent, which will deepen our understanding of the mechanism related to the treatment target of $\alpha 7$ nAChR. It also provides inspiration for the introduction of new therapies related to AKI. However, there are some defects in the design and results of this study, and some problems need to be solved before possible publication.

1. In general, there are many grammatical errors throughout the manuscript. Also, the authors should pay attentions to the proper usage of terminologies. I recommend they have some scholars fluent in English to thoroughly revise the manuscript and polish the language.

2. The author should describe each small figure of Fig.2, while the "Both anti inflammatory and renoprotective effects of GTS-21 are lost in macro specific $\alpha 7$ nAChR knockout mice" section only mentions Fig 2b,c,d; in addition, why do Fig.2f ($p=0.81$) and Fig.1f ($p=0.44$) have different results?

3. In Fig 5. although GTS-21 has been proved to promote the migration of macrophages through the transwell, this does not mean that there is communication between macrophages.

4. The current research in Fig.6 is too preliminary. To illustrate the impact of GTS-21 on the changes of macrophage anti-inflammatory phenotype, the marker level of corresponding phenotype (e.g. Arg-1, iNOS, etc.) should be detected simultaneously. In addition, flow cytometry is also a good choice to detect the phenotypic polarization of macrophages.

5. In addition to splenectomy, $\alpha 7$ nAChR silencing can further prove that it is a key factor to enhance the interaction between macrophages.

6. The molecular basis of CAP activation is the stimulation of $\alpha 7$ nAChR by endogenous acetylcholine. Therefore, during the treatment of GTS-21, on the one hand, the author should pay attention to the levels of ACh, choline acetyltransferase and so on to reflect the function of $\alpha 7$ nAChR; On the other hand, the expression level of $\alpha 7$ nAChR should also be detected.

7. Note the writing specifications in this document, e.g. RAW264.7 cells are not RAW264 cells. Please further check the full text.

Reviewer #4 (Remarks to the Author):

The manuscript by Nakamura et al. is incomplete. Validation of the macrophage-specific $\alpha 7$ nAChR KO mice is lacking. Without this, in my opinion, this paper must go back to the authors for revision. They must show the macrophage-specific knockout and the retention of $\alpha 7$ in other tissues. This may not be easy. Antibodies for $\alpha 7$ are notoriously unreliable, and they will not be able to use electrophysiological methods since $\alpha 7$ receptors in immune cells are not capable of generating ion currents.

The authors should also demonstrate a better knowledge of the drug they are relying on. They provide no reference to substantiate that GTS-21 is alpha7 selective or what its actual efficacy is. In fact, it has different efficacy in rats and humans, and I don't think it has been evaluated directly with mouse alpha7. It has strong desensitizing effects on alpha7 and is also an antagonist of other nAChR as well as 5HT3 receptors. Since it has been indicated that the alpha7-mediated activity in CAP may rely on receptors in non-conducting (i.e. desensitized conformations), the desensitizing effects of GTS-21 may be precisely why it appears to work so well at modulating inflammation. It might also be noted that in the in vivo experiments, GTS-21 may function as a pro-drug since its metabolites have better efficacy. Additionally, alpha9 receptors have also been implicated in CAP. What does GTS-21 do to them?

Page 8 Line 17: "No histological changes were observed following LPS administration with or without GTS-21 (Fig.1e and f)." This would seem to be exactly the opposite of what is shown in the +LPS, -GTS-21 data.

Page 7 line 13 needs a reference.

Response to Reviewers' comments/Questions

We appreciate the reviewers' comments and interest in our work.

Reviewer 1:

1. the cholinergic anti-inflammatory pathway through spleen macrophages is a "non-neuronal cholinergic pathway" by comparison to the one of the vagus nerve. Please specify.

Response: Thank you for your comment. We consider the mechanism of the cholinergic anti-inflammatory pathway (CAP) to be as follows.

The CAP was originally discovered as an anti-inflammatory pathway via cervical vagus nerve stimulation. This response was thought to occur because acetylcholine was released from efferent vagus nerve terminal fibers, thus inhibiting inflammatory cytokines (PMID: 10839541). The stimulation of the vagus nerve is known as the inflammatory reflex. Subsequent research has shown that in spleen macrophages, CAP exerts its anti-inflammatory effect by receiving acetylcholine released by ChAT-positive CD4⁺T cells in the spleen via $\alpha 7$ nAChR on macrophages (PMID: 28970585). As such, ACh is not received directly from the nerve terminals, but is considered to be a "non-neural cholinergic pathway" in that it occurs via immune cells, such as CD4⁺ T cells.

On the other hand, CAP that does not involve the spleen has also been reported in recent years. Especially when the vagus nerve directly innervates organs such as the lungs and intestines, ACh is released from the vagus nerve endings and is thought to have an organ-specific anti-inflammatory effect (PMID: 29331768).

We have added the following sentence to the introduction parts (page5, line17 to page6, line 7)

"This response in the spleen could be considered a "non-neuronal cholinergic pathway" in that ACh is not received directly from nerve endings but delivered via immune cells such as CD4⁺ T cells. One recently reported pathway does not involve the spleen and occurs, instead, in organs with vagal innervation, such as the lung and guts¹³. In these organs, ACh released from vagus nerve endings is thought to act directly on the organ and produce an anti-inflammatory effect. Moreover, since the parasympathetic innervation of the kidney has been controversial²⁹, it is currently believed that spleen-mediated CAP is the primary route of action for renal protection."

<"non-neural cholinergic pathway"> (ref PMID: 28970585)

<"not splenic" cholinergic anti-inflammatory pathway> (ref PMID: 29331768)

2. the authors did not explore the effect of sympathetic/vagal (splenic) denervation in their model.

Response: Thank you for your comment. In the concept of CAP, signals of the vagus nerve entering the spleen from splenic nerve terminals are transmitted to $\alpha 7nAChRs$ on macrophages via $CD4^+$ cells. Splenic nerves have been thought to be sympathetic, but Guyot et al. have proposed that certain parts of the splenic nerves are parasympathetic (PMID:30885844). Therefore, we performed experiments with $CD4^+$ T cells depletion instead of sympathetic/vagal (splenic) denervation (New Figure 7). Based on the concept of CAP, when $CD4^+$ T cells are depleted, acetylcholine is not released from T cells, and the administration of GTS-21 should not result in decreased production of $TNF-\alpha$ from macrophages. However, the results (Fig. 7d in the revised manuscript) show a decrease in $TNF-\alpha$ after administration of GTS-21 even after depleting $CD4^+$ T cells (Supplementary Fig. 10), supporting our hypothesis that macrophage-macrophage interaction in the spleen boosts $TNF-\alpha$ production.

Figure 7

Supplementary Figure 10

3. limitations of the study: the authors observed a large variability due to a wide range of age groups and sexes of mice (page 19, line 4). They had better to select these groups for their study.

Response: Thank you very much for your comment. We agree with your suggestion. We repeated the experiment again, with LysMCre: $\alpha 7$ flox mice of the same age and gender (male, 8-12 weeks old) (Supplementary Fig.3).

Even when these characteristics were strictly matched, variation in the results did not change significantly.

We hope you will find our revision adequate.

Supplementary Figure 3

Reviewer 2:

Major

1. Figure 2: GTS-21 does not seem to be reproductively inducing anti-inflammatory effects, making it difficult to truly state that CAP is relayed by $\alpha 7nAChR+$ macrophages: In contrast to the WT mice of Figure 1, the littermate WT of Figure 2 namely did not have reduced BUN and KIM levels following GTS-21 administration. Similar observations were made in Supplementary Figure 1.

In addition, both in Figure 1 and 2 GTS-21 did not reduce tubular injury score. Thus, the authors are overstating that GTS-21 has renoprotective effects. Maybe better to show that GTS-21 reduces apoptosis following LPS as showed by Gao et al (2017).

Response: We gratefully appreciate this suggestion. As you have noted, we did not find significant differences in BUN and Kim-1 in former Figure 2 and former Supplementary Figure 1 in the littermate WT group. Histologically assessed tubular injury scores also failed to detect the early damage caused by LPS administration. Following your advice, we conducted TUNEL staining with WT and LysMCre: $\alpha 7lox$ mice. TUNEL staining showed a decrease in apoptotic cells with GTS-treatment, so we have added these results to the main figures. (Figure 1. e and f and Figure2. e and f).

For reference, the results of Gao et al. were obtained using higher dose (LPS 10mg/kg vs 5mg/kg in our experiment) and longer exposure of LPS (16 h vs 4 h in our experiment). So there more severe conditions by Gao et al. showed more apoptotic cells than ours.

Figure 1

Figure 2

2. Since there are so few macrophages sequenced, the authors should not state that GTS-21 does not induce changes in the macrophage transcriptome. In fact, they should repeat the scRNA seq and FACS-sort to enrich for macrophages.

Response: We thank you for this comment. As you pointed out, there are very few macrophages in the spleen itself, so we enriched macrophages in the spleen using MACS and repeated the scRNA-seq. The results are shown in new figure 5 (Fig.5) of the revised manuscript.

Newly sequenced analysis data shows that GTS induces the expressions of genes related with focal adhesion (new Figure 5c-e). In addition, the data shows the expression of target genes such as integrin (*Itgal*) and macrophage regulatory factor (*Nr1h3*) were also predicted to be induced as a result of ligand-receptor interactions between macrophages and macrophages. It was also predicted to weaken the M1-induced macrophage phenotype (pro-inflammatory) by GTS-21 as shown in new Figure 5g. We are grateful for your suggestions that have helped make our hypothesis more convincing.

Figure 5

3. In addition, they do not show that these macrophages express $\alpha 7$ nAChR or other nicotinic receptors.

Response: We appreciate your insightful comments. We agree that this is an important point. We tried to confirm the expression of acetylcholine receptors including $\alpha 7$ nAChR in macrophages in scRNA-seq samples, but the expression levels were too low for us to detect Differentially Expressed Genes (DEGs). Further, databases (e.g. Mouse Cell Atlas <https://bis.zju.edu.cn/MCA/>, PanglaoDB <https://panglaoDB.se/index.html>) did not confirm $\alpha 7$ nAChR in splenocytes.

The expression of $\alpha 7$ nAChR in peritoneal macrophages extracted from mice and in RAW 264 cells was also examined, but could not be amplified using RT-PCR due to low expression levels. Similarly, protein expression (anti-CHRNA7, proteintech, #21379-1-AP/ anti-AChR $\alpha 7$, Santa Cruz, #sc-58607/ anti-CHRNA7, Bioss, #bs-1049R/ anti-CHRNA7, alomone labs, #ANC-007) was confirmed using Western blotting but was difficult to detect due to the lack of commercially available and reliable antibodies.

Although it was difficult to detect the expression of $\alpha 7$ nAChR directly, we confirmed that the expression of downstream factors of $\alpha 7$ nAChR is GTS-21 induced. We previously identified Hes-1 (hairy and enhancer of split 1) as a variable factor in nicotine-treated macrophages and reported that it is a downstream factor of CAP (PMID: 30670317). Hes-1 expression is also elevated by GTS-21 administration. This is indirect evidence, but it is possible that the downstream factor Hes1 is upregulated via $\alpha 7$ nAChR. Furthermore, to investigate the significance of endogenous acetylcholine receptors in vivo, we generated knockout mice (LysMCre: $\alpha 7$ flox mice) and confirmed that $\alpha 7$ nAChR could be deleted in macrophages (Supplementary Fig.2). We hope you will agree with our revision.

Supplementary Figure 2

4. I also miss some volcano plots showing differential expressed genes of macrophages and some SCENIC analysis to unravel to transcription factors leading to the macrophage phenotype in the spleen.

Response: Following your valuable advice, we have added a volcano plot for several genes induced by GTS-21 to new Fig.5 in the revised manuscript.

Furthermore, we have attempted SCENIC analysis.

The result of SCENIC analysis is shown below, but some of the factors that changed dynamically with the administration of GTS-21, so we have focused, instead, on Differentially Expressed Genes (DEGs) in macrophage-enriched samples and added the analysis results from NicheNet (<https://github.com/saeyslab/nichenetr>) shown in Fig. 5.

5. Figure 4: The authors ignore the fact that DC and NK cells have increased interactions with macrophages between LPS-Veh vs LPS-GTS. In addition, they do not make a deep characterization of macrophage phenotype following the different treatments. They only describe how many LR pairs are different, but do not discuss the genes until the discussion.

Response: As you have noted, the results obtained in this study showed increased interactions between DC, NK and NKT cells and macrophages.

For DC and NKT cells, the results of LR analysis with macrophages are shown below. We focused only on macrophages this time.

1. The source of TNF- α is macrophages.
2. Based on the results of macrophage-specific $\alpha 7$ AChR knockout mice (Fig.2), we focused on the interactions between macrophages and other immune cells that received signals through $\alpha 7$ AChR. While the interactions between DC or NKT cells and macrophages increased, but in many cases, the expression changes were observed on the ligand side (the side that transmits signals through the $\alpha 7$ AChR), and there were almost no changes in the interactions on the receptor side (the side that receives signals from macrophages that received $\alpha 7$ AChR signals).

This time, we focused on macrophage-macrophage because the source of TNF is macrophages. We did not mention these reasons in the original version, so I added these reasons to the discussion in the text as follows (Page 20, Line 7-10).

“Here, we focused on macrophage-macrophage interactions, considering that macrophage ligand expression levels must be subject to GTS-21-induced changes for $\alpha 7$ nAChR-signaling macrophages to interact and that macrophages are the main source of TNF- α .”

Additionally, the newly added NicheNet results show that GTS-21 suppresses macrophage phenotypes that are inclined to be inflammatory by LPS administration (new Fig.5h in the revised manuscript). We have added these outcomes to the discussions section (Page 20, line17-Page 21, line1).

“Indeed, Itgal was included in the DEGs upregulated by GTS-21 (Fig. 5d and e). We also found that GTS-21 might weaken the phenotype of macrophages that were changed pro-inflammatory (CD80, CD86) by LPS (Fig. 5h).”

6. Figure 5: Even though the authors state the GTS-21 increased cell-cell contact based on the transwell experiments, it just appears that GTS-21 increases migration. What factor do the author think that is inducing this transmigration?

Response: Thank you for your question. In our original study, we found that spleen scRNA-seq increased LR pairs between macrophages and macrophages, and that cell contact increased in a transwell migration assay, but we could not show what factors might be involved.

In the present study, scRNA-seq analysis of macrophage-enriched samples revealed that GTS-21 treatment affected the expression of *Itga4* (Integrin Subunit Alpha 4) and *Csf3* (colony stimulating factor 3) ligands in one cluster of macrophages and *Itgal* (which is involved in cell adhesion in other clusters of macrophages) in the other cluster of macrophages. It was found that ligands such as *Itga4* (Integrin Subunit Alpha 4) and *Csf3* (colony stimulating factor 3) present in one cluster of macrophages increased the expression of *Itgal* (Integrin Alpha-L), which is involved in cell adhesion in other clusters of macrophages (Fig.5f). In fact, focal adhesion was found to increase after treatment with GTS-21(Fig.5c). It is likely that these splenic macrophage interactions promote cell adhesion.

We added following sentence to discussion part (Page20, line12 to page 20, line18)

“Furthermore, predictions of ligand-receptor communication affected by GTS-21 by NicheNet in macrophage-enriched samples (Fig. 5f-h) predicted that macrophages receiving GTS-21 expressed *Itga4* (Integrin Subunit Alpha 4) and *Csf3* (colony stimulating factor 3) ligands and predicted to induce expression of target genes such as *Itgal* (Integrin Alpha-L) and *Nr1h3* (genes involved in the regulation of macrophage function) in other cluster macrophages (Fig. 5f). Indeed, *Itgal* was included in the DEGs upregulated by GTS-21 (Fig. 5d and e).”

We hope you will agree with our revision here.

7. Figure 6: Do the authors obtain similar results if they just double the concentration of RAW cells from 6×10^4 cells to 12×10^4 ?

Response: We agree with you. We had the same concern and have performed the following experiments (Supplementary Fig.7). We performed the same experiment with RAW cells (6×10^4) instead of U937 (c) and compared the results with a dish with twice as many RAW cells (12×10^4) (d). Since the number of RAW cells in the dishes in (c) and (d) are the same, the amount of TNF- α produced is expected to be the same. When we seeded 12×10^4 RAW cells in dish (d), the amount of TNF- α produced doubled. However, when we added new

RAW cells (6×10^4) to the dish that originally contained 6×10^4 cells (c), the amount of TNF- α produced decreased by under 1.5 times. This suggests that there might have been a decrease in the production of TNF- α due to interactions between the RAW cells originally seeded in the dish and the newly added RAW cells.

Supplemental Figure 7

8. Figure 7: The authors did splenectomy to determine if cell-cell interactions between macrophages in the spleen is essential for the CAP. It however seems that this experiment just proves the spleen is required for the CAP, not specifically the macrophages.

Response: Thank you for your comment. As you pointed out, splenectomy experiment alone is insufficient to demonstrate the role of macrophages interactions.

However, since it is difficult to experimentally prove cell-cell interactions in vivo, we have added experiments supporting cell-cell interaction in the spleen.

We have re-done scRNA-seq in macrophage-enriched samples and found that gene

expression related with focal adhesion between macrophage-macrophages is increased (Fig.5c). Furthermore, we demonstrated that GTS-21 increases macrophage migration and decreases TNF- α production when macrophages are in contact with each other in an in vitro transwell migration assay (Fig. 6). We also confirmed that α -bungarotoxin (α -Batx), an α 7nAChR antagonist, attenuates macrophage migration (Supplementary Fig.6). Taken together, these results suggest that α 7nAChR signaling and the cell-cell interaction of macrophages is critical for exerting an anti-inflammatory effect and our experiment using splenectomy supports this hypothesis.

Figure 5

Figure 6

Figure 6

f

g

Supplementary Figure 6

a

b

d

c

e

Minor

Instead of research papers, a lot of review papers are cited in the introduction

Instead of 9, 10: VNS in Crohn: 26920654, 32515156; colitis: 34307422 31133776

Instead of 58  23929694

Response: Thank you for your remarks. We have replaced some of those references.

- 9 Sinniger, V. *et al.* A 12-month pilot study outcomes of vagus nerve stimulation in Crohn's disease. *Neurogastroenterology & Motility* **32**, 1-16 (2020).
- 10 Meroni, E. *et al.* Vagus Nerve Stimulation Promotes Epithelial Proliferation and Controls Colon Monocyte Infiltration During DSS-Induced Colitis. *Front Med (Lausanne)* **8**, 694268 (2021).
- 11 Bonaz, B. *et al.* Chronic vagus nerve stimulation in Crohn's disease: a 6-month follow-up pilot study. *Neurogastroenterology & Motility* **28**, 948-953 (2016).
- 12 Payne, S. C. *et al.* Anti-inflammatory Effects of Abdominal Vagus Nerve Stimulation on Experimental Intestinal Inflammation. *Front Neurosci* **13**, 418 (2019).
- 63 Matteoli, G. *et al.* A distinct vagal anti-inflammatory pathway modulates intestinal muscularis resident macrophages independent of the spleen. *Gut* **63**, 938-948 (2014).

Reviewer 3.

1. In general, there are many grammatical errors throughout the manuscript. Also, the authors should pay attentions to the proper usage of terminologies. I recommend they have some scholars fluent in English to thoroughly revise the manuscript and polish the language.

Response: Thank you very much for this suggestion. We have revised the manuscript for language, focusing on our choice of words and vocabulary. We have also had the manuscript proofread by a native speaker by acquiring language-editing services from Editage (<https://www.editage.jp/>).

2. The author should describe each small figure of Fig.2, while the "Both anti inflammatory and renoprotective effects of GTS-21 are lost in macro specific $\alpha 7nAChR$ knockout mice" section only mentions Fig 2b,c,d; in addition, why do Fig.2f ($p=0.81$) and Fig.1f ($p=0.44$) have different results?

Response: Thank you for your careful review of our work. In our previous experiments, we did not observe a significant histological difference in tubular injury scores at the early stages of injury, 4 hours after LPS administration, so we found no significant differences in Fig2f ($p=0.81$) and Fig1f ($p=0.44$). To detect early damage by LPS, we performed TUNEL staining to count apoptotic cells and found GTS-21 decreased the number of apoptotic cells even 4 hours after LPS administration. Therefore, we added the TUNEL staining results to the main figure (Fig.1 e, f and Fig.2 e, f) and moved the histological tubular injury score to the supplementary figures (Supple Fig.1 and Fig.4).

Figure 1

Figure 2

3. In Fig 5. although GTS-21 has been proved to promote the migration of macrophages through the transwell, this does not mean that there is communication between macrophages.

Response: We agree with you that it would be an overstatement to say that the transwell migration assay alone increased macrophage interactions.

In the transwell migration assay alone (previous Fig. 5), only macrophage migration was observed. However, when combined with the result that increased macrophage-macrophage contact decreases TNF- α production (previous Fig. 6), we considered that in vitro, we have also observed an anti-inflammatory effect due to increased macrophage contact rather than a direct GTS-21 effect (new Fig. 6 containing the previous Fig.5 and Fig.6). We have also proved that macrophage contacts were decreased by α -bungarotoxin (α -Btx), an α 7nAChR antagonist (Supplementary Fig.6). The results of scRNA-seq using macrophage-enriched samples (Fig.5c) show that the increased expression of adhesion factors imply the increased contact with macrophages, but further investigation is needed. We hope you will agree with our revision.

Figure 6

Figure 6

Figure 5

Supplementary Figure 6

4. The current research in Fig.6 is too preliminary. To illustrate the impact of GTS-21 on the changes of macrophage anti-inflammatory phenotype, the marker level of corresponding phenotype (e.g. Arg-1, iNOS, etc.) should be detected simultaneously. In addition, flow cytometry is also a good choice to detect the phenotypic polarization of macrophages.

Response: Thank you for your suggestion. We conducted RT-PCR to check the phenotypic change by treatment with GTS-21 using RAW 264 cells (Supplementary Fig.9). Although this is not statistically significant, one can see the decreasing trend in iNos and an increasing trend in Arg1 were observed with GTS administration.

No statistically significant differences were observed, possibly because 4 hours is too short a time to capture the changes in macrophage phenotype.

In addition, inflammatory cytokines such as IL1 β and IL-6 are suppressed even 4 hours after GTS-21 treatment, supporting that GTS-21 alters the phenotype of macrophages. (Supplementary Fig.8).

Supplementary Figure 9

a

b

Supplementary Figure 8

5. In addition to splenectomy, $\alpha 7$ nAChR silencing can further prove that it is a key factor to enhance the interaction between macrophages.

Response: Thank you for your great suggestion. To silence $\alpha 7$ nAChR, we performed a transwell migration assay using α -bungarotoxin (α -Batx), a specific antagonist of $\alpha 7$ nAChR (Supplemental Fig.6). α -Batx administration decreased the migration of RAW cells, which increased when the agonist GTS was administered. These results support our hypothesis that GTS acts to increase macrophage-macrophage interactions.

Supplementary Figure 6

6. The molecular basis of CAP activation is the stimulation of α 7nAChR by endogenous acetylcholine. Therefore, during the treatment of GTS-21, on the one hand, the author should pay attention to the levels of ACh, choline acetyltransferase and so on to reflect the function of α 7nAChR; On the other hand, the expression level of α 7nAChR should also be detected.

Response: Thank you for your comments. Acetylcholine is difficult to measure because it is quickly degraded in the absence of AChE. Therefore, we instead used RT-PCR to confirm the expression of choline acetyltransferase (ChAT), an acetylcholine synthase. ChAT in macrophages increases with the administration of GTS-21 as shown below.

We previously identified Hes-1 (hairy and enhancer of split 1) as a variable factor in nicotine-treated macrophages and reported that it is a downstream factor of CAP (PMID: 30670317). Hes-1 expression is also elevated by GTS-21 administration, suggesting that GTS-21 regulates CAP via α 7nAChR.

a**b**
7. Note the writing specifications in this document, e.g. RAW264.7 cells are not RAW264 cells. Please further check the full text.

Response: We thank for your careful review of our work. The RAW cells used in our experiments were purchased from the RIKEN BRC CELL BANK, and the cell name is RAW 264 (RCB0535; https://cellbank.brc.riken.jp/cell_bank/CellInfo/?cellNo=RCB0535). The notation in the text has been checked and standardized.

Reviewer 4.

1. The manuscript by Nakamura et al. is incomplete. Validation of the macrophage-specific alpha7 nAChR KO mice is lacking. Without this, in my opinion, this paper must go back to the authors for revision. They must show the macrophage-specific knockout and the retention of alpha7 in other tissues. This may not be easy. Antibodies for alpha7 are notoriously unreliable, and they will not be able to use electrophysiological methods since alpha7 receptors in immune cells are not capable of generating ion currents.

Response: Thank you for your advice. As you pointed out, there are no reliable antibodies for $\alpha 7$ nAChR, and the RNA expression of *chrna7* ($\alpha 7$ nAChR) in the tissue is very low, so it was difficult to prove the expression of $\alpha 7$ nAChR using RT-PCR.

We used genotyping to confirm the traits, so we added those results to Supplemental Fig.2b. To reinforce the genotyping results, we also created primer pairs designed to detect exon 4 of $\alpha 7$ nAChR (*Chrna7*; Supplementary Fig.2a) and measured DNA levels in macrophages using peritoneal macrophages and other tissues (tail; Supplementary Fig.2c). Supplemental Fig.2c confirms that exon4 of *Chrna7* is deleted in macrophages.

Supplementary Figure 2

2. The authors should also demonstrate a better knowledge of the drug they are relying on. They provide no reference to substantiate that GTS-21 is $\alpha 7$ selective or what its actual efficacy is. In fact, it has different efficacy in rats and humans, and I don't think it has been evaluated directly with mouse $\alpha 7$. It has strong desensitizing effects on $\alpha 7$ and is also an antagonist of other nAChR as well as 5HT₃ receptors. Since it has been indicated that the $\alpha 7$ -mediated activity in CAP may rely on receptors in non-conducting (i.e. desensitized conformations), the desensitizing effects of GTS-21 may be precisely why it appears to work so well at modulating inflammation. It might also be noted that in the in vivo experiments, GTS-21 may function as a pro-drug since its metabolites have better efficacy. Additionally, $\alpha 9$ receptors have also been implicated in CAP. What does GTS-21 do to them?

Response: Thank you for your feedback. Your comments have given us valuable information about GTS-21 that was previously missing from our manuscript. As you pointed out, GTS-21 is known as a partial agonist for the $\alpha 7$ and $\alpha 4\beta 2$ subtypes of nicotinic acetylcholine receptors. In particular, GTS-21 activates only $\alpha 7$ nAChR to a potent extent in rats (PMID: 9164577, PMID: 9369300). The validation of GTS-21 in mice by Moriguchi et al. demonstrated that chronic nicotine exposure with PNU-282987 (specific agonists of $\alpha 7$ nAChR) and GTS-21 reduce depressive behavior in calcium/calmodulin-dependent protein kinase IV (CaMKIV) null mice and that this response is mediated by $\alpha 7$, not $\alpha 4\beta 2$ subtypes (PMID: 32815115). Indeed, there are several reports that GTS-21 exerts anti-inflammatory effects via CAP in mice and reduces kidney injury (PMID:18401335, 22586448, 29190774, 31105861, 35311383, 36572735).

As you pointed out, it has been shown that $\alpha 7$ subunit might act as a silent agonist for inflammatory pain and CAP (PMID: 31874200).

Furthermore, it is reported that $\alpha 9$ subunit is activated in the low μ M range, whereas $\alpha 7$ activation requires agonist stimulation in the high mM range in vitro (PMID: 11752216). Based on these findings, we cannot rule out the possibility that $\alpha 9$ is also stimulated by GTS-21, but our results using macrophage-specific $\alpha 7$ subtype knockout mice in vivo may provide evidence for the involvement of the $\alpha 7$ subunit in CAP.

Based on the excellent review by Hone et al. (PMID: 36868367) and Papke et al. (PMID: 36940890), we have added information about GTS-21 to the discussion sections (page23, line15-Page 24, line1). Once more, we deeply appreciate your valuable suggestion for discussing CAP in further detail.

"In addition, it should be noted that GTS-21 used in this study is an agonist not only for the

$\alpha 7$ nicotinic receptor but also for the $\alpha 4\beta 2$ subtype and has been reported to act as a silent agonist^{69,70}. Recently, it has been reported that the $\alpha 9$ subunit nicotinic acetylcholine receptor is involved in the activation of CAP^{71,72}, and the verification of the role of $\alpha 9$ may be necessary for the future.”

3. Page 8 Line 17: "No histological changes were observed following LPS administration with or without GTS-21 (Fig.1e and f)." This would seem to be exactly the opposite of what is shown in the +LPS, -GTS-21 data.

Response: Thank you for your careful check. As you pointed out, in the original Fig.1e and f, the histological injury induced by LPS was not significantly different, although there was a tendency for a slight decrease with the administration of GTS-21. We think that the histological injury was mild at 4 hours after LPS administration, which was the condition of this experiment, and it was difficult to compare the degree of damage with and without GTS-21. Instead, we evaluated apoptosis by TUNEL staining, and found a significant difference in apoptosis reduction by GTS-21, so we have replaced the main figure (Fig.1e and f, Fig.2e and f).

4. Page 7 line 13 needs a reference.

Response: Thank you for your comments. We have added the following reference.

- 55 Clausen, B. E., Burkhardt, C., Reith, W., Renkawitz, R. & Förster, I. Conditional gene targeting in macrophages and granulocytes using LysMcre mice. *Transgenic Research* **8**, 265-277 (1999).
- 56 Hernandez, C. M. *et al.* Research tool: Validation of floxed $\alpha 7$ nicotinic acetylcholine receptor conditional knockout mice using in vitro and in vivo approaches. *J Physiol* **592**, 3201-3214 (2014).

Response to Reviewers' comments/Questions

We appreciate the reviewers' comments and interest in our work.

Reviewer 1:

1. the cholinergic anti-inflammatory pathway through spleen macrophages is a "non-neuronal cholinergic pathway" by comparison to the one of the vagus nerve. Please specify.

Response: Thank you for your comment. We consider the mechanism of the cholinergic anti-inflammatory pathway (CAP) to be as follows.

The CAP was originally discovered as an anti-inflammatory pathway via cervical vagus nerve stimulation. This response was thought to occur because acetylcholine was released from efferent vagus nerve terminal fibers, thus inhibiting inflammatory cytokines (PMID: 10839541). The stimulation of the vagus nerve is known as the inflammatory reflex. Subsequent research has shown that in spleen macrophages, CAP exerts its anti-inflammatory effect by receiving acetylcholine released by ChAT-positive CD4⁺T cells in the spleen via $\alpha 7$ nAChR on macrophages (PMID: 28970585). As such, ACh is not received directly from the nerve terminals, but is considered to be a "non-neural cholinergic pathway" in that it occurs via immune cells, such as CD4⁺ T cells.

On the other hand, CAP that does not involve the spleen has also been reported in recent years. Especially when the vagus nerve directly innervates organs such as the lungs and intestines, ACh is released from the vagus nerve endings and is thought to have an organ-specific anti-inflammatory effect (PMID: 29331768).

We have added the following sentence to the introduction parts (page5, line17 to page6, line 7)

"This response in the spleen could be considered a "non-neuronal cholinergic pathway" in that ACh is not received directly from nerve endings but delivered via immune cells such as CD4⁺ T cells. One recently reported pathway does not involve the spleen and occurs, instead, in organs with vagal innervation, such as the lung and guts¹³. In these organs, ACh released from vagus nerve endings is thought to act directly on the organ and produce an anti-inflammatory effect. Moreover, since the parasympathetic innervation of the kidney has been controversial²⁹, it is currently believed that spleen-mediated CAP is the primary route of action for renal protection."

<"non-neural cholinergic pathway"> (ref PMID: 28970585)

<"not splenic" cholinergic anti-inflammatory pathway> (ref PMID: 29331768)

2. the authors did not explore the effect of sympathetic/vagal (splenic) denervation in their model.

Response: Thank you for your comment. In the concept of CAP, signals of the vagus nerve entering the spleen from splenic nerve terminals are transmitted to $\alpha 7nAChRs$ on macrophages via $CD4^+$ cells. Splenic nerves have been thought to be sympathetic, but Guyot et al. have proposed that certain parts of the splenic nerves are parasympathetic (PMID:30885844). Therefore, we performed experiments with $CD4^+$ T cells depletion instead of sympathetic/vagal (splenic) denervation (New Figure 7). Based on the concept of CAP, when $CD4^+$ T cells are depleted, acetylcholine is not released from T cells, and the administration of GTS-21 should not result in decreased production of $TNF-\alpha$ from macrophages. However, the results (Fig. 7d in the revised manuscript) show a decrease in $TNF-\alpha$ after administration of GTS-21 even after depleting $CD4^+$ T cells (Supplementary Fig. 10), supporting our hypothesis that macrophage-macrophage interaction in the spleen boosts $TNF-\alpha$ production.

Figure 7

Supplementary Figure 10

3. limitations of the study: the authors observed a large variability due to a wide range of age groups and sexes of mice (page 19, line 4). They had better to select these groups for their study.

Response: Thank you very much for your comment. We agree with your suggestion. We repeated the experiment again, with LysMCre: $\alpha 7$ flox mice of the same age and gender (male, 8-12 weeks old) (Supplementary Fig.3).

Even when these characteristics were strictly matched, variation in the results did not change significantly.

We hope you will find our revision adequate.

Supplementary Figure 3

Reviewer 2:

Major

1. Figure 2: GTS-21 does not seem to be reproductively inducing anti-inflammatory effects, making it difficult to truly state that CAP is relayed by $\alpha 7nAChR+$ macrophages: In contrast to the WT mice of Figure 1, the littermate WT of Figure 2 namely did not have reduced BUN and KIM levels following GTS-21 administration. Similar observations were made in Supplementary Figure 1.

In addition, both in Figure 1 and 2 GTS-21 did not reduce tubular injury score. Thus, the authors are overstating that GTS-21 has renoprotective effects. Maybe better to show that GTS-21 reduces apoptosis following LPS as showed by Gao et al (2017).

Response: We gratefully appreciate this suggestion. As you have noted, we did not find significant differences in BUN and Kim-1 in former Figure 2 and former Supplementary Figure 1 in the littermate WT group. Histologically assessed tubular injury scores also failed to detect the early damage caused by LPS administration. Following your advice, we conducted TUNEL staining with WT and LysMCre: $\alpha 7lox$ mice. TUNEL staining showed a decrease in apoptotic cells with GTS-treatment, so we have added these results to the main figures. (Figure 1. e and f and Figure2. e and f).

For reference, the results of Gao et al. were obtained using higher dose (LPS 10mg/kg vs 5mg/kg in our experiment) and longer exposure of LPS (16 h vs 4 h in our experiment). So there more severe conditions by Gao et al. showed more apoptotic cells than ours.

Figure 1

Figure 2

2. Since there are so few macrophages sequenced, the authors should not state that GTS-21 does not induce changes in the macrophage transcriptome. In fact, they should repeat the scRNA seq and FACS-sort to enrich for macrophages.

Response: We thank you for this comment. As you pointed out, there are very few macrophages in the spleen itself, so we enriched macrophages in the spleen using MACS and repeated the scRNA-seq. The results are shown in new figure 5 (Fig.5) of the revised manuscript.

Newly sequenced analysis data shows that GTS induces the expressions of genes related with focal adhesion (new Figure 5c-e). In addition, the data shows the expression of target genes such as integrin (*Itgal*) and macrophage regulatory factor (*Nr1h3*) were also predicted to be induced as a result of ligand-receptor interactions between macrophages and macrophages. It was also predicted to weaken the M1-induced macrophage phenotype (pro-inflammatory) by GTS-21 as shown in new Figure 5g. We are grateful for your suggestions that have helped make our hypothesis more convincing.

Figure 5

3. In addition, they do not show that these macrophages express $\alpha 7$ nAChR or other nicotinic receptors.

Response: We appreciate your insightful comments. We agree that this is an important point. We tried to confirm the expression of acetylcholine receptors including $\alpha 7$ nAChR in macrophages in scRNA-seq samples, but the expression levels were too low for us to detect Differentially Expressed Genes (DEGs). Further, databases (e.g. Mouse Cell Atlas <https://bis.zju.edu.cn/MCA/>, PanglaoDB <https://panglaoDB.se/index.html>) did not confirm $\alpha 7$ nAChR in splenocytes.

The expression of $\alpha 7$ nAChR in peritoneal macrophages extracted from mice and in RAW 264 cells was also examined, but could not be amplified using RT-PCR due to low expression levels. Similarly, protein expression (anti-CHRNA7, proteintech, #21379-1-AP/ anti-AChR $\alpha 7$, Santa Cruz, #sc-58607/ anti-CHRNA7, Bioss, #bs-1049R/ anti-CHRNA7, alomone labs, #ANC-007) was confirmed using Western blotting but was difficult to detect due to the lack of commercially available and reliable antibodies.

Although it was difficult to detect the expression of $\alpha 7$ nAChR directly, we confirmed that the expression of downstream factors of $\alpha 7$ nAChR is GTS-21 induced. We previously identified Hes-1 (hairy and enhancer of split 1) as a variable factor in nicotine-treated macrophages and reported that it is a downstream factor of CAP (PMID: 30670317). Hes-1 expression is also elevated by GTS-21 administration. This is indirect evidence, but it is possible that the downstream factor Hes1 is upregulated via $\alpha 7$ nAChR. Furthermore, to investigate the significance of endogenous acetylcholine receptors in vivo, we generated knockout mice (LysMCre: $\alpha 7$ flox mice) and confirmed that $\alpha 7$ nAChR could be deleted in macrophages (Supplementary Fig.2). We hope you will agree with our revision.

Supplementary Figure 2

4. I also miss some volcano plots showing differential expressed genes of macrophages and some SCENIC analysis to unravel to transcription factors leading to the macrophage phenotype in the spleen.

Response: Following your valuable advice, we have added a volcano plot for several genes induced by GTS-21 to new Fig.5 in the revised manuscript.

Furthermore, we have attempted SCENIC analysis.

The result of SCENIC analysis is shown below, but some of the factors that changed dynamically with the administration of GTS-21, so we have focused, instead, on Differentially Expressed Genes (DEGs) in macrophage-enriched samples and added the analysis results from NicheNet (<https://github.com/saeyslab/nichenetr>) shown in Fig. 5.

5. Figure 4: The authors ignore the fact that DC and NK cells have increased interactions with macrophages between LPS-Veh vs LPS-GTS. In addition, they do not make a deep characterization of macrophage phenotype following the different treatments. They only describe how many LR pairs are different, but do not discuss the genes until the discussion.

Response: As you have noted, the results obtained in this study showed increased interactions between DC, NK and NKT cells and macrophages.

For DC and NKT cells, the results of LR analysis with macrophages are shown below. We focused only on macrophages this time.

1. The source of TNF- α is macrophages.
2. Based on the results of macrophage-specific $\alpha 7$ AChR knockout mice (Fig.2), we focused on the interactions between macrophages and other immune cells that received signals through $\alpha 7$ AChR. While the interactions between DC or NKT cells and macrophages increased, but in many cases, the expression changes were observed on the ligand side (the side that transmits signals through the $\alpha 7$ AChR), and there were almost no changes in the interactions on the receptor side (the side that receives signals from macrophages that received $\alpha 7$ AChR signals).

This time, we focused on macrophage-macrophage because the source of TNF is macrophages. We did not mention these reasons in the original version, so I added these reasons to the discussion in the text as follows (Page 20, Line 7-10).

“Here, we focused on macrophage-macrophage interactions, considering that macrophage ligand expression levels must be subject to GTS-21-induced changes for $\alpha 7$ nAChR-signaling macrophages to interact and that macrophages are the main source of TNF- α .”

Additionally, the newly added NicheNet results show that GTS-21 suppresses macrophage phenotypes that are inclined to be inflammatory by LPS administration (new Fig.5h in the revised manuscript). We have added these outcomes to the discussions section (Page 20, line17-Page 21, line1).

“Indeed, Itgal was included in the DEGs upregulated by GTS-21 (Fig. 5d and e). We also found that GTS-21 might weaken the phenotype of macrophages that were changed pro-inflammatory (CD80, CD86) by LPS (Fig. 5h).”

6. Figure 5: Even though the authors state the GTS-21 increased cell-cell contact based on the transwell experiments, it just appears that GTS-21 increases migration. What factor do the author think that is inducing this transmigration?

Response: Thank you for your question. In our original study, we found that spleen scRNA-seq increased LR pairs between macrophages and macrophages, and that cell contact increased in a transwell migration assay, but we could not show what factors might be involved.

In the present study, scRNA-seq analysis of macrophage-enriched samples revealed that GTS-21 treatment affected the expression of *Itga4* (Integrin Subunit Alpha 4) and *Csf3* (colony stimulating factor 3) ligands in one cluster of macrophages and *Itgal* (which is involved in cell adhesion in other clusters of macrophages) in the other cluster of macrophages. It was found that ligands such as *Itga4* (Integrin Subunit Alpha 4) and *Csf3* (colony stimulating factor 3) present in one cluster of macrophages increased the expression of *Itgal* (Integrin Alpha-L), which is involved in cell adhesion in other clusters of macrophages (Fig.5f). In fact, focal adhesion was found to increase after treatment with GTS-21 (Fig.5c). It is likely that these splenic macrophage interactions promote cell adhesion.

We added following sentence to discussion part (Page20, line12 to page 20, line18)

“Furthermore, predictions of ligand-receptor communication affected by GTS-21 by NicheNet in macrophage-enriched samples (Fig. 5f-h) predicted that macrophages receiving GTS-21 expressed *Itga4* (Integrin Subunit Alpha 4) and *Csf3* (colony stimulating factor 3) ligands and predicted to induce expression of target genes such as *Itgal* (Integrin Alpha-L) and *Nr1h3* (genes involved in the regulation of macrophage function) in other cluster macrophages (Fig. 5f). Indeed, *Itgal* was included in the DEGs upregulated by GTS-21 (Fig. 5d and e).”

We hope you will agree with our revision here.

7. Figure 6: Do the authors obtain similar results if they just double the concentration of RAW cells from 6×10^4 cells to 12×10^4 ?

Response: We agree with you. We had the same concern and have performed the following experiments (Supplementary Fig.7). We performed the same experiment with RAW cells (6×10^4) instead of U937 (c) and compared the results with a dish with twice as many RAW cells (12×10^4) (d). Since the number of RAW cells in the dishes in (c) and (d) are the same, the amount of TNF- α produced is expected to be the same. When we seeded 12×10^4 RAW cells in dish (d), the amount of TNF- α produced doubled. However, when we added new

RAW cells (6×10^4) to the dish that originally contained 6×10^4 cells (c), the amount of TNF- α produced decreased by under 1.5 times. This suggests that there might have been a decrease in the production of TNF- α due to interactions between the RAW cells originally seeded in the dish and the newly added RAW cells.

Supplemental Figure 7

8. Figure 7: The authors did splenectomy to determine if cell-cell interactions between macrophages in the spleen is essential for the CAP. It however seems that this experiment just proves the spleen is required for the CAP, not specifically the macrophages.

Response: Thank you for your comment. As you pointed out, splenectomy experiment alone is insufficient to demonstrate the role of macrophages interactions.

However, since it is difficult to experimentally prove cell-cell interactions in vivo, we have added experiments supporting cell-cell interaction in the spleen.

We have re-done scRNA-seq in macrophage-enriched samples and found that gene

expression related with focal adhesion between macrophage-macrophages is increased (Fig.5c). Furthermore, we demonstrated that GTS-21 increases macrophage migration and decreases TNF- α production when macrophages are in contact with each other in an in vitro transwell migration assay (Fig. 6). We also confirmed that α -bungarotoxin (α -Batx), an α 7nAChR antagonist, attenuates macrophage migration (Supplementary Fig.6). Taken together, these results suggest that α 7nAChR signaling and the cell-cell interaction of macrophages is critical for exerting an anti-inflammatory effect and our experiment using splenectomy supports this hypothesis.

Figure 5

Figure 6

Figure 6

f

g

Supplementary Figure 6

a

b

d

c

e

Minor

Instead of research papers, a lot of review papers are cited in the introduction

Instead of 9, 10: VNS in Crohn: 26920654, 32515156; colitis: 34307422 31133776

Instead of 58  23929694

Response: Thank you for your remarks. We have replaced some of those references.

- 9 Sinniger, V. *et al.* A 12-month pilot study outcomes of vagus nerve stimulation in Crohn's disease. *Neurogastroenterology & Motility* **32**, 1-16 (2020).
- 10 Meroni, E. *et al.* Vagus Nerve Stimulation Promotes Epithelial Proliferation and Controls Colon Monocyte Infiltration During DSS-Induced Colitis. *Front Med (Lausanne)* **8**, 694268 (2021).
- 11 Bonaz, B. *et al.* Chronic vagus nerve stimulation in Crohn's disease: a 6-month follow-up pilot study. *Neurogastroenterology & Motility* **28**, 948-953 (2016).
- 12 Payne, S. C. *et al.* Anti-inflammatory Effects of Abdominal Vagus Nerve Stimulation on Experimental Intestinal Inflammation. *Front Neurosci* **13**, 418 (2019).
- 63 Matteoli, G. *et al.* A distinct vagal anti-inflammatory pathway modulates intestinal muscularis resident macrophages independent of the spleen. *Gut* **63**, 938-948 (2014).

Reviewer 3.

1. In general, there are many grammatical errors throughout the manuscript. Also, the authors should pay attentions to the proper usage of terminologies. I recommend they have some scholars fluent in English to thoroughly revise the manuscript and polish the language.

Response: Thank you very much for this suggestion. We have revised the manuscript for language, focusing on our choice of words and vocabulary. We have also had the manuscript proofread by a native speaker by acquiring language-editing services from Editage (<https://www.editage.jp/>).

2. The author should describe each small figure of Fig.2, while the "Both anti inflammatory and renoprotective effects of GTS-21 are lost in macro specific $\alpha 7nAChR$ knockout mice" section only mentions Fig 2b,c,d; in addition, why do Fig.2f ($p=0.81$) and Fig.1f ($p=0.44$) have different results?

Response: Thank you for your careful review of our work. In our previous experiments, we did not observe a significant histological difference in tubular injury scores at the early stages of injury, 4 hours after LPS administration, so we found no significant differences in Fig2f ($p=0.81$) and Fig1f ($p=0.44$). To detect early damage by LPS, we performed TUNEL staining to count apoptotic cells and found GTS-21 decreased the number of apoptotic cells even 4 hours after LPS administration. Therefore, we added the TUNEL staining results to the main figure (Fig.1 e, f and Fig.2 e, f) and moved the histological tubular injury score to the supplementary figures (Supple Fig.1 and Fig.4).

Figure 1

Figure 2

3. In Fig 5. although GTS-21 has been proved to promote the migration of macrophages through the transwell, this does not mean that there is communication between macrophages.

Response: We agree with you that it would be an overstatement to say that the transwell migration assay alone increased macrophage interactions.

In the transwell migration assay alone (previous Fig. 5), only macrophage migration was observed. However, when combined with the result that increased macrophage-macrophage contact decreases TNF- α production (previous Fig. 6), we considered that in vitro, we have also observed an anti-inflammatory effect due to increased macrophage contact rather than a direct GTS-21 effect (new Fig. 6 containing the previous Fig.5 and Fig.6). We have also proved that macrophage contacts were decreased by α -bungarotoxin (α -Btx), an α 7nAChR antagonist (Supplementary Fig.6). The results of scRNA-seq using macrophage-enriched samples (Fig.5c) show that the increased expression of adhesion factors imply the increased contact with macrophages, but further investigation is needed. We hope you will agree with our revision.

Figure 6

Figure 6

Figure 5

Supplementary Figure 6

4. The current research in Fig.6 is too preliminary. To illustrate the impact of GTS-21 on the changes of macrophage anti-inflammatory phenotype, the marker level of corresponding phenotype (e.g. Arg-1, iNOS, etc.) should be detected simultaneously. In addition, flow cytometry is also a good choice to detect the phenotypic polarization of macrophages.

Response: Thank you for your suggestion. We conducted RT-PCR to check the phenotypic change by treatment with GTS-21 using RAW 264 cells (Supplementary Fig.9). Although this is not statistically significant, one can see the decreasing trend in iNos and an increasing trend in Arg1 were observed with GTS administration.

No statistically significant differences were observed, possibly because 4 hours is too short a time to capture the changes in macrophage phenotype.

In addition, inflammatory cytokines such as IL1 β and IL-6 are suppressed even 4 hours after GTS-21 treatment, supporting that GTS-21 alters the phenotype of macrophages. (Supplementary Fig.8).

Supplementary Figure 9

a

b

Supplementary Figure 8

5. In addition to splenectomy, $\alpha 7$ nAChR silencing can further prove that it is a key factor to enhance the interaction between macrophages.

Response: Thank you for your great suggestion. To silence $\alpha 7$ nAChR, we performed a transwell migration assay using α -bungarotoxin (α -Batx), a specific antagonist of $\alpha 7$ nAChR (Supplemental Fig.6). α -Batx administration decreased the migration of RAW cells, which increased when the agonist GTS was administered. These results support our hypothesis that GTS acts to increase macrophage-macrophage interactions.

Supplementary Figure 6

6. The molecular basis of CAP activation is the stimulation of α 7nAChR by endogenous acetylcholine. Therefore, during the treatment of GTS-21, on the one hand, the author should pay attention to the levels of ACh, choline acetyltransferase and so on to reflect the function of α 7nAChR; On the other hand, the expression level of α 7nAChR should also be detected.

Response: Thank you for your comments. Acetylcholine is difficult to measure because it is quickly degraded in the absence of AChE. Therefore, we instead used RT-PCR to confirm the expression of choline acetyltransferase (ChAT), an acetylcholine synthase. ChAT in macrophages increases with the administration of GTS-21 as shown below.

We previously identified Hes-1 (hairy and enhancer of split 1) as a variable factor in nicotine-treated macrophages and reported that it is a downstream factor of CAP (PMID: 30670317). Hes-1 expression is also elevated by GTS-21 administration, suggesting that GTS-21 regulates CAP via α 7nAChR.

a**b**
7. Note the writing specifications in this document, e.g. RAW264.7 cells are not RAW264 cells. Please further check the full text.

Response: We thank for your careful review of our work. The RAW cells used in our experiments were purchased from the RIKEN BRC CELL BANK, and the cell name is RAW 264 (RCB0535; https://cellbank.brc.riken.jp/cell_bank/CellInfo/?cellNo=RCB0535). The notation in the text has been checked and standardized.

Reviewer 4.

1. The manuscript by Nakamura et al. is incomplete. Validation of the macrophage-specific alpha7 nAChR KO mice is lacking. Without this, in my opinion, this paper must go back to the authors for revision. They must show the macrophage-specific knockout and the retention of alpha7 in other tissues. This may not be easy. Antibodies for alpha7 are notoriously unreliable, and they will not be able to use electrophysiological methods since alpha7 receptors in immune cells are not capable of generating ion currents.

Response: Thank you for your advice. As you pointed out, there are no reliable antibodies for $\alpha 7$ nAChR, and the RNA expression of *chrna7* ($\alpha 7$ nAChR) in the tissue is very low, so it was difficult to prove the expression of $\alpha 7$ nAChR using RT-PCR.

We used genotyping to confirm the traits, so we added those results to Supplemental Fig.2b. To reinforce the genotyping results, we also created primer pairs designed to detect exon 4 of $\alpha 7$ nAChR (*Chrna7*; Supplementary Fig.2a) and measured DNA levels in macrophages using peritoneal macrophages and other tissues (tail; Supplementary Fig.2c). Supplemental Fig.2c confirms that exon4 of *Chrna7* is deleted in macrophages.

Supplementary Figure 2

2. The authors should also demonstrate a better knowledge of the drug they are relying on. They provide no reference to substantiate that GTS-21 is $\alpha 7$ selective or what its actual efficacy is. In fact, it has different efficacy in rats and humans, and I don't think it has been evaluated directly with mouse $\alpha 7$. It has strong desensitizing effects on $\alpha 7$ and is also an antagonist of other nAChR as well as 5HT₃ receptors. Since it has been indicated that the $\alpha 7$ -mediated activity in CAP may rely on receptors in non-conducting (i.e. desensitized conformations), the desensitizing effects of GTS-21 may be precisely why it appears to work so well at modulating inflammation. It might also be noted that in the in vivo experiments, GTS-21 may function as a pro-drug since its metabolites have better efficacy. Additionally, $\alpha 9$ receptors have also been implicated in CAP. What does GTS-21 do to them?

Response: Thank you for your feedback. Your comments have given us valuable information about GTS-21 that was previously missing from our manuscript. As you pointed out, GTS-21 is known as a partial agonist for the $\alpha 7$ and $\alpha 4\beta 2$ subtypes of nicotinic acetylcholine receptors. In particular, GTS-21 activates only $\alpha 7$ nAChR to a potent extent in rats (PMID: 9164577, PMID: 9369300). The validation of GTS-21 in mice by Moriguchi et al. demonstrated that chronic nicotine exposure with PNU-282987 (specific agonists of $\alpha 7$ nAChR) and GTS-21 reduce depressive behavior in calcium/calmodulin-dependent protein kinase IV (CaMKIV) null mice and that this response is mediated by $\alpha 7$, not $\alpha 4\beta 2$ subtypes (PMID: 32815115). Indeed, there are several reports that GTS-21 exerts anti-inflammatory effects via CAP in mice and reduces kidney injury (PMID:18401335, 22586448, 29190774, 31105861, 35311383, 36572735).

As you pointed out, it has been shown that $\alpha 7$ subunit might act as a silent agonist for inflammatory pain and CAP (PMID: 31874200).

Furthermore, it is reported that $\alpha 9$ subunit is activated in the low μ M range, whereas $\alpha 7$ activation requires agonist stimulation in the high mM range in vitro (PMID: 11752216). Based on these findings, we cannot rule out the possibility that $\alpha 9$ is also stimulated by GTS-21, but our results using macrophage-specific $\alpha 7$ subtype knockout mice in vivo may provide evidence for the involvement of the $\alpha 7$ subunit in CAP.

Based on the excellent review by Hone et al. (PMID: 36868367) and Papke et al. (PMID: 36940890), we have added information about GTS-21 to the discussion sections (page23, line15-Page 24, line1). Once more, we deeply appreciate your valuable suggestion for discussing CAP in further detail.

"In addition, it should be noted that GTS-21 used in this study is an agonist not only for the

$\alpha 7$ nicotinic receptor but also for the $\alpha 4\beta 2$ subtype and has been reported to act as a silent agonist^{69,70}. Recently, it has been reported that the $\alpha 9$ subunit nicotinic acetylcholine receptor is involved in the activation of CAP^{71,72}, and the verification of the role of $\alpha 9$ may be necessary for the future.”

3. Page 8 Line 17: "No histological changes were observed following LPS administration with or without GTS-21 (Fig.1e and f)." This would seem to be exactly the opposite of what is shown in the +LPS, -GTS-21 data.

Response: Thank you for your careful check. As you pointed out, in the original Fig.1e and f, the histological injury induced by LPS was not significantly different, although there was a tendency for a slight decrease with the administration of GTS-21. We think that the histological injury was mild at 4 hours after LPS administration, which was the condition of this experiment, and it was difficult to compare the degree of damage with and without GTS-21. Instead, we evaluated apoptosis by TUNEL staining, and found a significant difference in apoptosis reduction by GTS-21, so we have replaced the main figure (Fig.1e and f, Fig.2e and f).

4. Page 7 line 13 needs a reference.

Response: Thank you for your comments. We have added the following reference.

- 55 Clausen, B. E., Burkhardt, C., Reith, W., Renkawitz, R. & Förster, I. Conditional gene targeting in macrophages and granulocytes using LysMcre mice. *Transgenic Research* **8**, 265-277 (1999).
- 56 Hernandez, C. M. *et al.* Research tool: Validation of floxed $\alpha 7$ nicotinic acetylcholine receptor conditional knockout mice using in vitro and in vivo approaches. *J Physiol* **592**, 3201-3214 (2014).